# Visualizing chaperone-mediated multistep assembly of the human 20S proteasome

Frank Adolf [1,2,7] ✉, Jiale Du [1,2,7], Ellen A. Goodall [2,3], Richard M. Walsh Jr [4,5], Shaun Rawson [4,5], Susanne von Gronau[1], J. Wade Harper [2,3], John Hanna [6] ✉ & Brenda A. Schulman [1,2] ✉

Dedicated assembly factors orchestrate the stepwise production of many molecular machines, including the 28-subunit proteasome core particle (CP) that mediates protein degradation. Here we report cryo-electron microscopy reconstructions of seven recombinant human subcomplexes that visualize all five chaperones and the three active site propeptides across a wide swath of the assembly pathway. Comparison of these chaperone-bound intermediates and a matching mature CP reveals molecular mechanisms determining the order of successive subunit additions, as well as how proteasome subcomplexes and assembly factors structurally adapt upon progressive subunit incorporation to stabilize intermediates, facilitate the formation of subsequent intermediates and ultimately rearrange to coordinate proteolytic activation with gated access to active sites. This work establishes a methodologic approach for structural analysis of multiprotein complex assembly intermediates, illuminates specific functions of assembly factors and reveals conceptual principles underlying human proteasome biogenesis, thus providing an explanation for many previous biochemical and genetic observations.

Numerous cellular functions are performed by large multisubunit complexes. However, because of their size and complexity, many complexes cannot assemble spontaneously. Instead, they are built through ordered multistep pathways orchestrated by dedicated chaperones and assembly factors, which are excluded from the mature complex. One example is the proteasome, which mediates regulated intracellular protein degradation and degrades misfolded or aggregation-prone proteins associated with neurodegenerative and other diseases.

The proteasome's proteolytic active sites are centrally located inside its 700-kDa barrel-shaped core particle (CP). The CP is composed of four stacked α-rings and β-rings, each containing seven distinct α or β subunits[1,2]. The active sites are sequestered within the CP interior and are, therefore, accessible only through a narrow-gated pore formed by the α-ring at either end[3].

Unlike conventional proteases, the proteasome harbors three different types of active site (β1 or caspase-like, β2 or trypsin-like and β5 or chymotrypsin-like), each present twice[2,4,5]. This arrangement allows the proteasome to be both comprehensive in its substrate repertoire and processive. The proteasome's exceptional substrate specificity is not inherent in the CP but is provided through regulators that bind the α-ring to open the CP gates and facilitate substrate entry.

[1]Department of Molecular Machines and Signaling, Max Planck Institute of Biochemistry, Martinsried, Germany. [2]Aligning Science Across Parkinson's (ASAP) Collaborative Research Network, Chevy Chase, MD, USA. [3]Department of Cell Biology, Harvard Medical School, Boston, MA, USA. [4]Harvard Cryo-Electron Microscopy Center for Structural Biology, Harvard Medical School, Boston, MA, USA. [5]Department of Biological Chemistry and Molecular Pharmacology, Blavatnik Institute, Harvard Medical School, Boston, MA, USA. [6]Department of Pathology, Harvard Medical School and Brigham and Women's Hospital, Boston, MA, USA. [7]These authors contributed equally: Frank Adolf, Jiale Du. ✉e-mail: fadolf@biochem.mpg.de; jwhanna@bwh.harvard.edu; schulman@biochem.mpg.de

The best-characterized CP regulator is the 19S regulatory particle (RP), which recognizes ubiquitylated substrates[6]. Meanwhile, various PA28 isoforms form homohexamers or heterohexamers that enable the immunoproteasome to process peptides for antigen presentation[7–9]. Gate opening is achieved through C-terminal HbYX motifs (Hb, hydrophobic; Y, tyrosine; X, any residue) present in various CP regulators, which insert into the pockets between adjacent α subunits[10–17]. PA200 also possesses an HbYX motif, although its cellular role remains unclear[18–21].

Approximately 3,000 CPs are generated per minute in proliferating HeLa cells (copy number ~$4 \times 10^6$)[22], with quantities increased in response to stress[23]. Given the abundance and complexity of CPs, it stands to reason that proteasome biogenesis depends on dedicated chaperones that facilitate assembly but are not subunits of mature proteasomes[24–28]. Previous work identified five CP chaperones: proteasome maturation protein (POMP, also called hUmp1)[29–34] and proteasome assembly chaperones 1–4 (PAC1–PAC4)[35–38]. Moreover, five of the seven β subunits (β1, β2 and β5–β7) are synthesized as inactive precursors with N-terminal propeptides, some of which contribute to CP biogenesis[39–42]. The importance of proper proteasome biogenesis is underscored by a group of human diseases caused by variations in these chaperones or regions of CP subunits critical for assembly[43–45], and by the early embryonic lethality observed in mice with knockout of *Dusp2* (also known as *Pac1*)[46].

CP biogenesis is thought to begin with the formation of a complete α-ring in a process that requires PAC1–PAC2 (hereafter PAC1/PAC2) and PAC3–PAC4 (hereafter PAC3/PAC4)[35–37,47–51], with the two heterodimers bound to opposite sides of the α-ring. Next, the β subunits are thought to incorporate sequentially onto the α-ring[42], ultimately resulting in a half-CP. Two half-CPs then fuse to create the complete CP barrel. However, the CP remains inactive until cleavage of the β subunit propeptides, which occurs through a poorly understood autocatalytic mechanism for the active site subunits. Once activation occurs, POMP is destroyed, and PAC1/PAC2 binding is also eliminated, resulting in mature, active CP[35,52,53].

In contrast to PAC1/PAC2, PAC3/PAC4 are released early during CP biogenesis, although the timing and trigger for release have remained unclear. PAC3/PAC4 are not present in the best-characterized intermediate, the 13S complex, which has been identified even in wild-type mammalian cells and consists of PAC1/PAC2, the complete α-ring and β2–β4 (refs. 33,39,54,55). Although no structural information is yet available for PAC3/PAC4-containing intermediates, modeling of the corresponding yeast α5–Pba3–Pba4 (hereafter Pba3/Pba4) crystal structure onto a mature CP suggested steric clash between Pba3/Pba4 and the β4 subunit, suggesting that insertion of β4 might induce release of Pba3/Pba4 (ref. 47). Alternatively, knockdown of β3 in human cells resulted in accumulation of an intermediate containing all five chaperones[42], raising the possibility that PAC3/PAC4 release is coordinated with insertion of β3, despite the lack of obvious steric clash. Yet another possibility was suggested by steric clash between Pba3/Pba4 and Ump1 in high-resolution structures of yeast 13S and pre-15S intermediates[56].

In addition to the uncertainty regarding PAC3/PAC4, other major aspects of CP biogenesis remain poorly understood. These include the mechanisms underlying the regimented stepwise production of the β-ring and the coupling of CP assembly with peptidase activation[16,40,52,53,57]. Here, we used cryo-electron microscopy (cryo-EM) to obtain and characterize structures of human assembly chaperone-containing proteasome CP subcomplexes. We report the structure for the PAC3/PAC4-containing CP intermediate and a series of structures containing increasing numbers of β subunits, yielding a detailed understanding of the stepwise assembly of the β-ring. This study greatly advances our understanding of proteasome assembly, with implications for the broader field of chaperone-mediated multiprotein complex biogenesis.

## Results

### Recombinant human CP expression

Recombinant systems have enabled the study of mature proteasome complexes[20,58–60]. To express human CP in insect cells, we used a combination of biGBac[61] and MultiBac[62] systems to generate baculovirus transfer vectors: one with all seven α subunits, another with all seven β subunits and a third with the five CP assembly chaperones. For affinity purification, we appended a C-terminal twin-Strep tag to either β2 or β7, the first and last CP subunits, respectively, to incorporate. We performed affinity purification on CP, followed by size-exclusion chromatography (SEC) (Fig. 1a,b). Both preparations showed the expected electrophoretic properties for mature CP. Furthermore, cryo-EM analyses of the peak fractions from both preparations yielded high-resolution structures that superimposed with native human 20S CP (root-mean-square deviation (r.m.s.d.) 0.703 Å) (Fig. 1c, right, and Extended Data Figs. 1a–e and 2).

The SEC profile for the β2-tag purification revealed a shoulder peak consistent with lower-molecular-weight subcomplexes (Fig. 1a). SDS–PAGE analysis revealed additional proteins with a size of 12–15 kDa. Some assembly chaperones migrate in this size range, suggesting that these complexes might represent immature CP species. Cryo-EM analysis of these fractions yielded structures of five distinct immature CP subcomplexes with overall resolutions ranging from 2.67 to 2.95 Å (Fig. 1c, left, Extended Data Figs. 3 and 4 and Table 1). We order these structures on the basis of the number of visible β subunits, with their compositions indicated in Fig. 1c. We present the structures with α subunits in blue and β subunits in purple, increasing in shade by subunit number (α-ring) or order of incorporation (β-ring). These structures are consistent with the ordered, sequential addition of β subunits proposed from genetic knockdown studies in human cells[42].

Structure 1 represents an early CP subcomplex and consists of the α-ring, all five chaperones and β2 (Fig. 1c and Supplementary Video 1). Structures 2–5 consist of the α-ring, chaperones POMP and PAC1/PAC2, and increasing numbers of β subunits (Fig. 1c and Supplementary Videos 2–5). In addition to mature CP, detailed analysis of the cryo-EM data from the higher-molecular-weight fraction revealed two supra-20S structures, referred to as 'preholo 20S CP' and 'premature 20S CP' (Fig. 1c, right, and Extended Data Fig. 1d–f). Structures 3 and 4 are analogous to recent structures of yeast 13S and pre-15S intermediates, respectively[56]. Meanwhile, cryo-EM maps of various resolutions have also shown constituents of yeast preholo 20S and premature 20S CP complexes[63,64]. Thus, the structural progression of human complexes lies within a conserved proteasome assembly pathway. Comparing the seven different structures to each other and to matched mature CP allows visualizing the conformational progression across the assembly pathway (Supplementary Video 6) and the specific molecular interactions occurring at each step, as described below.

### Chaperones coordinate α-ring formation and β-ring initiation

Structure 1 shows PAC1/PAC2 and PAC3/PAC4 within the same complex and illustrates POMP-dependent initiation of β-ring assembly through its binding of both the α-ring and β2 (Supplementary Video 1). Together, PAC1/PAC2 and PAC3/PAC4 bind all seven α subunits (Fig. 2a,b, zoomed-out views). PAC1/PAC2 is perched atop the α-ring (Fig. 2a), on the surface recognized by proteasome activators. This placement appears to serve two main purposes. First, it likely helps coordinate α-ring configuration because PAC1/PAC2 contact all α subunits except α3 (Fig. 2c). Second, as observed for the yeast counterpart Pba1/Pba2 in later proteasome assembly intermediates[27], PAC1/PAC2 directly bind the N termini from multiple α subunits (Fig. 2a, close-up) and, through numerous interactions, orchestrate an open-gate conformation (Fig. 2d–g). Perhaps most dramatically, PAC1/PAC2 bind α5's N terminus more than 20 Å away from the α-ring (Fig. 2a and Supplementary Video 1). This interaction likely contributes to PAC1/PAC2's reported ability to bind α5 in isolation[35].

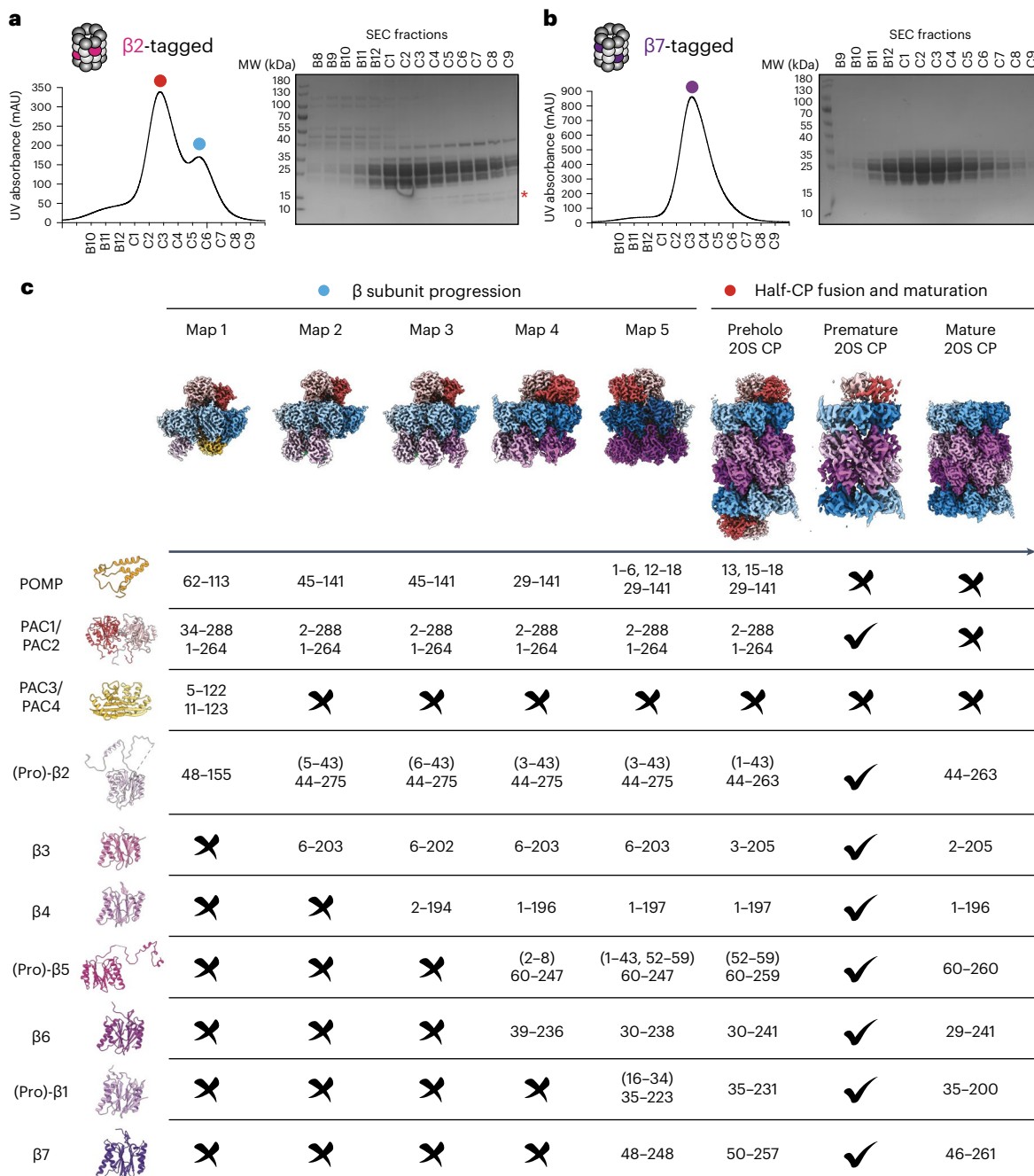

**Fig. 1 | Recombinant expression of human 20S proteasome in insect cells.**
**a**, Purification of recombinant β2-tagged (PSMB7) 20S CP. Eluates were subjected to SEC, and the fractions were analyzed by SDS–PAGE, followed by Coomassie staining. The asterisk indicates bands in the later fractions (corresponding to complexes smaller than a mature 20S CP) of lower molecular weight than proteasome subunits that would correspond to the assembly chaperones PAC1–PAC4 and POMP. Red, blue and purple dots indicate fractions used for cryo-EM, with the later-migrating fraction (blue dot) yielding high-resolution structures 1–5 of 20S CP assembly intermediates and the early fraction (red dot)

yielding structures of β2-tagged preholo, premature and mature 20S CP. MW, molecular weight; mAU, milli-absorbance units. **b**, Purification of recombinant β7-tagged (PSMB4) 20S CP. The single-peak fraction (purple dot) yielded a high-resolution structure of β7-tagged 20S CP. **c**, Cryo-EM maps of eight distinct CP subcomplexes. Their subunits, the chaperone configurations and the resolved region in each map are indicated. Check marks and cross marks indicate subunits that are visible or absent in the corresponding maps, respectively. Similar results for **a** and **b** were obtained in three independent experiments.

The visualized N termini from α subunits have features in common with those in substrate-engaged 26S proteasome[65] (Supplementary Video 7), which shows an open gate. This arrangement sharply contrasts with the mature CP, wherein the N termini from α subunits extend into and block access to the central pore[2]. Similar to known proteasome activators such as the RP (Fig. 2d)[12,15,19–21,66], PAC1 contains a canonical HbYX motif (Ile-Tyr-Thr)[48,56], which inserts into the α5–α6

pocket (Fig. 2e). PAC2 contains a variant sequence (Leu-Phe), with Phe positioned in the α6–α7 pocket similarly to conventional interactions with the Tyr residue from the HbYX motif (Fig. 2f).

PAC3/PAC4 sit on the opposite face of the α-ring (that is, the future site of β4–β7 subunits later during assembly) (Fig. 2a,b and Supplementary Videos 1 and 6). Consistent with the structure of the orthologous isolated yeast Pba3/Pba4–α5 complex (Extended Data Fig. 5a)[47,67,68],

**Table 1 | Cryo-EM data collection, refinement and validation statistics**

| Map | 1 | 2 | 3 | 4 | 5 | 5[87] | Preholo 20S CP | Premature 20S CP | 20S CP | 20S CP[87] |
|---|---|---|---|---|---|---|---|---|---|---|
| **Data acquisition** | | | | | | | | | | |
| PDB | 8QYJ | 8QYL | 8QYM | 8QZ9 | 8QYN | – | 8QYS | – | 8QYO | – |
| EMDB | EMD-18755 | EMD-18757 | EMD-18758 | EMD-18773 | EMD-18759 | EMD-19342 | EMD-18761 | EMD-18762 | EMD-18760 | EMD-19343 |
| **Data collection** | | | | | | | | | | |
| Microscope | TFS Titan Krios | TFS Titan Krios | TFS Titan Krios | TFS Titan Krios | TFS Titan Krios | TFS Glacios | TFS Glacios | TFS Glacios | TFS Glacios | TFS Glacios |
| Voltage (kV) | 300 | 300 | 300 | 300 | 300 | 200 | 200 | 200 | 200 | 200 |
| Exposure navigation | Stage movement and beamshift | Stage movement and beamshift | Stage movement and beamshift | Stage movement and beamshift | Stage movement and beamshift | Stage movement and beamshift | Stage movement and beamshift | Stage movement and beamshift | Stage movement and beamshift | Stage movement and beamshift |
| Automation software | SerialEM | SerialEM | SerialEM | SerialEM | SerialEM | SerialEM | SerialEM | SerialEM | SerialEM | SerialEM |
| Detector | Gatan K3 | Gatan K3 | Gatan K3 | Gatan K3 | Gatan K3 | Gatan K2 Summit | Gatan K2 Summit | Gatan K2 Summit | Gatan K2 Summit | Gatan K2 Summit |
| Energy filter | 10 eV | 10 eV | 10 eV | 10 eV | 10 eV | None | None | None | None | None |
| Nominal magnification | ×105,000 | ×105,000 | ×105,000 | ×105,000 | ×105,000 | ×36,000 | ×22,000 | ×22,000 | ×36,000 | ×36,000 |
| Pixel size (Å) | 0.8512 | 0.8512 | 0.8512 | 0.8512 | 0.8512 | 1.181 | 1.885 | 1.885 | 1.181 | 1.181 |
| Exposure time (s), frames | 3/30 | 3/30 | 3/30 | 3/30 | 3/30 | 17.2/40 | 16/40 | 16/40 | 14/40 | 17.2/40 |
| Exposure rate ($e^-$ $Å^{-2}$ $s^{-1}$) | 22.30 | 22.30 | 22.30 | 22.30 | 22.30 | 3.48 | 3.75 | 3.75 | 4.025 | 3.48 |
| Electron exposure ($e^-$ $Å^{-2}$) | 66.90 | 66.90 | 66.90 | 66.90 | 66.90 | 60.00 | 60.00 | 60.00 | 56.35 | 60.00 |
| Defocus range (µm) | –1.0 to –2.6 | –1.0 to –2.6 | –1.0 to –2.6 | –1.0 to –2.6 | –1.0 to –2.6 | –1.0 to –2.6 | –1.0 to –2.6 | –1.0 to –2.6 | –1.0 to –2.6 | –1.0 to –2.6 |
| Micrographs collected | 19,310 | 19,310 | 19,310 | 19,310 | 19,310 | 2,573 | 6,560 | 6,560 | 2,327 | 2,573 |
| **Reconstruction** | | | | | | | | | | |
| Software | cryoSPARC version 4.2 | cryoSPARC version 4.2 | cryoSPARC version 4.2 | cryoSPARC version 4.2 | cryoSPARC version 4.2 | cryoSPARC version 4.2 | cryoSPARC version 4.2 | cryoSPARC version 4.2 | cryoSPARC version 4.2 | cryoSPARC version 4.2 |
| Micrographs used | 18,006 | 18,006 | 18,006 | 18,006 | 18,006 | 2,171 | 4,429 | 4,429 | 1,804 | 2,171 |
| Particles used in refinement | 318,973 | 322,874 | 217,491 | 41,421 | 45,700 | 46,953 | 59,535 | 51,699 | 569,831 | 186,363 |
| Symmetry | C1 | C1 | C1 | C1 | C1 | C1 | C2 | C1 | C2 | C2 |
| Overall resolution (Å) gold-standard FSC (masked) | 2.73 | 2.67 | 2.73 | 2.95 | 2.88 | 4.11 | 3.89 | 3.93 | 2.84 | 3.13 |
| Map local resolution range (Å) | 2.38–6.09 | 1.84–5.88 | 1.84–6.13 | 1.83–7.41 | 1.83–7.41 | 3.63–9.36 | 3.89–7.43 | 3.93–8.97 | 2.645–4.99 | 2.78–6.38 |
| Map sharpening $B$ factor ($Å^2$) | –102.1 | –101.3 | –98.1 | –79.0 | –76.2 | –75 | –92.8 | –76.9 | –93.8 | –91.9 |
| **Model refinement** | | | | | | | | | | |
| Software | Phenix | Phenix | Phenix | Phenix | Phenix | – | Phenix | – | Phenix | – |
| Non-hydrogen atoms | 19,402 | 20,997 | 22,235 | 24,217 | 29,115 | – | 57,862 | – | 48,036 | – |
| Protein residues | 2,498 | 2,675 | 2,848 | 3,213 | 3,731 | – | 7,588 | – | 6,184 | – |
| Average $B$ factors ($Å^2$) | 63.37 | 82.89 | 67.25 | 99.85 | 92.38 | – | 85.22 | – | 48.57 | – |
| R.m.s.d. | | | | | | | | | | |
| Bond lengths (Å) | 0.002 | 0.003 | 0.002 | 0.003 | 0.004 | – | 0.003 | – | 0.005 | – |
| Bond angles (°) | 0.443 | 0.508 | 0.454 | 0.484 | 0.491 | – | 0.558 | – | 0.579 | – |
| Ramachandran outliers (%) | 0 | 0 | 0 | 0 | 0 | – | 0 | – | 0 | – |
| **Validation** | | | | | | | | | | |
| MolProbity score | 1.05 | 1.24 | 1.20 | 1.44 | 1.33 | – | 2.03 | – | 1.26 | – |
| Clashscore | 2.60 | 3.83 | 2.86 | 3.64 | 3.20 | – | 8.98 | – | 3.39 | – |
| Poor rotamers (%) | 0 | 0 | 0 | 0 | 0 | – | 0 | – | 0 | – |
| Model vs. map FSC = 0.143 (masked, Å) | 2.5 | 2.5 | 2.5 | 2.8 | 2.7 | – | 3.8 | – | 2.8 | – |
| Model local resolution range (Å) | 2.38–6.09 | 1.84–5.88 | 1.84–6.13 | 1.83–7.41 | 1.83–7.41 | 3.63–9.36 | 3.89–7.43 | 3.93–8.97 | 2.645–4.99 | 2.78–6.38 |

PAC3/PAC4 occupy the α3–α7 side of the ring, with α5 binding at the PAC3/PAC4 interface (Fig. 2b). By occupying these sites, PAC3/PAC4 not only stabilize the α-ring to provide a platform for β-ring assembly, but also preclude incorporation of the late β subunits, helping to order the incorporation of β subunits.

Unexpectedly, PAC3/PAC4 and the β-ring chaperone POMP directly interact (Fig. 2a,b). The same site in POMP bound by PAC3 is recognized later during assembly by the β5 propeptide (Extended Data Fig. 5b and Supplementary Videos 6 and 8). POMP consists of conserved helices and loops that become progressively more resolved upon the successive addition of β subunits, with POMP ultimately winding back and forth between the α-rings and β-rings (Extended Data Fig. 5c and Supplementary Video 8). In structure 1, one end of POMP is enclosed by PAC3, α7 and α1 (Fig. 2h), with the other end enclosed by PAC4, α3 and α4 (Fig. 2i). These contacts rationalize previous observations of POMP's direct interaction with the α-ring in isolation and with α3, α4 and α7 according to a yeast two-hybrid assay[42,69].

Lastly, structure 1 reveals β-ring initiation with β2; a PAC3/PAC4-bound complete α-ring engages POMP, which co-recruits β2 (Supplementary Video 1). The well-resolved portion of β2 is that inserted between α1 and α2 and in contact with POMP (Fig. 2h). The distal regions of β2 are poorly resolved, presumably due to flexibility without constraint from adjacent β subunits.

### The β2 propeptide and POMP secure β3 and β4

Structure 2 is distinguished by the loss of PAC3/PAC4 and addition of β3 (Fig. 2j,k and Supplementary Videos 2 and 6), and it explains previous observations from genetically engineered mammalian cells that β3 is the second β subunit to enter and this requires prior β2 addition[42]. The β2 subunit grasps β3 with both a C-terminal extension (CTE) and the β2 propeptide (Fig. 2j), rationalizing why β3 incorporation in mammalian cells requires these β2 elements[42]. Extending from the β2 active site Thr, the propeptide meanders, forms a helix crossing β3's interior surface and extends to secure newly visible regions of POMP that co-fasten β3 in the groove between α2 and α3 (Fig. 2j, bottom left). Meanwhile, part of β2's CTE forms a strand that extends a β-sheet from the adjacent β3, similar to the arrangement seen in mature CP[2,70].

β3 is stabilized by newly visible residues of POMP (residues 45–61 and 114–141) in structure 2 compared to structure 1 (Fig. 2j). Residues 114–141 directly clamp β3 against the α-ring through an interaction that is also buttressed by part of the β2 propeptide (Fig. 2j, bottom). POMP residues 45–61 buttress part of the β2 propeptide that secures the interior surface of β3 (Fig. 2j, bottom). Notably, the regions of the β2 propeptide and POMP that clamp β3 in place bind the same α3 and α4 surfaces as PAC3/PAC4 did in structure 1 (Fig. 2a,j, zoomed-out views, and Supplementary Video 6). This suggests that PAC3/PAC4 eviction may coincide with the stabilization of those regions of POMP and the β2 propeptide upon incorporation of β3. This could explain why β3 knockdown causes accumulation of the five-chaperone-containing intermediate[42] even though β3 does not directly overlap with PAC3/PAC4.

Upon departure of PAC3/PAC4, α-ring stability may be alternatively supported by POMP bridging β2–β3 and the α-ring, as well as changes at the α-ring pore. The N termini of α1 and PAC1 are newly visible in structure 2, passing through the gate toward POMP (Fig. 2k and Supplementary Video 2). PAC1's N terminus runs alongside α1's N terminus through the center of the α-ring until PAC1 contacts a newly ordered region of POMP and makes a U-turn to terminate in the center of the α-ring pore (Fig. 2k,l and Supplementary Video 2). Concomitantly, additional N-terminal residues of α2–α4 become visible, stabilized in an open conformation by a collar of Tyr residues between α-ring subunits rather than by direct, extensive interactions with PAC1/PAC2, as occurs for α5 (Extended Data Fig. 5d). This open-gate but occluded pore arrangement is maintained throughout the other assemblies we describe, including the preholo 20S CP.

Key differences between structures 1 and 2 set the stage for the β4 incorporation seen in structure 3 (Fig. 3a and Supplementary Video 3). One side of β4 binds a composite surface from β3 and the β2 propeptide, while its other side binds the α4 region that was bound to PAC4 in structure 1 (Fig. 2a,m and Supplementary Video 6). Importantly, structure 3 corresponds to the well-recognized 13S precursor[39,55] and is largely superimposable on yeast 13S[56], although there are notable differences accommodating the divergent PAC1 and yeast Pba1 N termini. First, while the N termini of both PAC1 and Pba1 thread through the open gate into the CP interior, in yeast, the N terminus of α2 (ref. 66), not α1, runs alongside Pba1's N terminus (Extended Data Fig. 5e,f). Second, PAC1's longer N terminus loops back upward toward the α-ring gate, whereas Pba1's N terminus ends in contact with Ump1 and the β5 propeptide[56] (Extended Data Fig. 5g).

### A POMP–propeptide network internally scaffolds the β-ring

Structure 4 corresponds to the yeast pre-15S intermediate[56] and contains the α-ring, β2–β6, PAC1/PAC2 and POMP, while structure 5 also contains β1 and β7 and shows better resolution of β6 (Supplementary Videos 4 and 5). These structures illustrate how portions of assembly factors become progressively structured as they secure the remaining β subunits (Supplementary Video 8 and Extended Data Figs. 6 and 7). Insertion of β5 and β6 is accompanied by increased visualization of POMP's N terminus (residues 29–44) (Fig. 3a,b, left) accompanied by redirection of the β2 propeptide's N terminus to fill a newly formed pocket at the confluence of POMP, β4–β5 and α3–α4 (Fig. 3a,b, right).

In structure 4, only the extreme N terminus (residues 2–8) of the β5 propeptide is visible, at the junction of α6, α7 and POMP. This resembles a harpoon, anchored 60 Å away from the rest of β5 (Fig. 3c,d and Supplementary Video 4). Structure 5 shows the assembly factors within a completed β-ring just before half-CP fusion (Supplementary Video 5). Additional elements in the β5 propeptide and POMP form a belt supporting the β5–β1 subunits (Fig. 4a,c). Broader regions of the β5 propeptide are newly visible in structure 5 compared to structure 4, suggesting that its folding is coordinated with the progressive addition of β subunits (Fig. 4a,b and Supplementary Videos 6 and 8). After emerging from mature β5, the β5 propeptide winds back and forth between the β-ring and α-ring to secure β6, β7 and β1 and then crosses POMP before terminating with the interactions already observed in structure 4 (Fig. 4b,c). The β5 propeptide's interactions at β6's interface with the α subunits clarify observations that the β5 propeptide is essential for the incorporation of β6, while β5 does not require its own propeptide for incorporation[42]. Meanwhile, newly visible N-terminal regions of POMP in structure 5 support the β6–β5 and β1–β2 pairs of subunits (Fig. 4b, right). A portion of the β1 propeptide (residues 16–34), visualized in structure 5, resembles a staple extending from β1 to β7 (Fig. 4d).

### Half-CP fusion and proteolytic activation

Comparing the half-CP in structure 5 and the preholo, premature and mature 20S CP structures reveals conformational changes accompanying half-CP fusion and eventual loss of propeptides, POMP and PAC1/PAC2 during 20S proteasome maturation (Fig. 5a and Supplementary Video 9). The most striking differences between structure 5 and preholo 20S CP are in the conformations of the β subunits. Fusion of two half-CPs coaxes the proteases toward active conformations (Fig. 5b–e). In particular, two key elements from each subunit along the β-ring–β-ring interface are a β-hairpin (for example, residues 61–75 of β2) and adjacent loop (for example, residues 205–215 of β2) (Fig. 5c). We refer to these as 'fusion hairpins' and 'fusion loops', respectively, because they interlock in a zipper-like structure with those from the opposing subunit across the β-ring–β-ring interface (Extended Data Fig. 8a–h). However, in the half-CP (structure 5), these elements are exposed, and either the majority are not visible and presumably dynamic (β2, β3, β5 and β1) or the β-hairpin is splayed outward or inward relative to

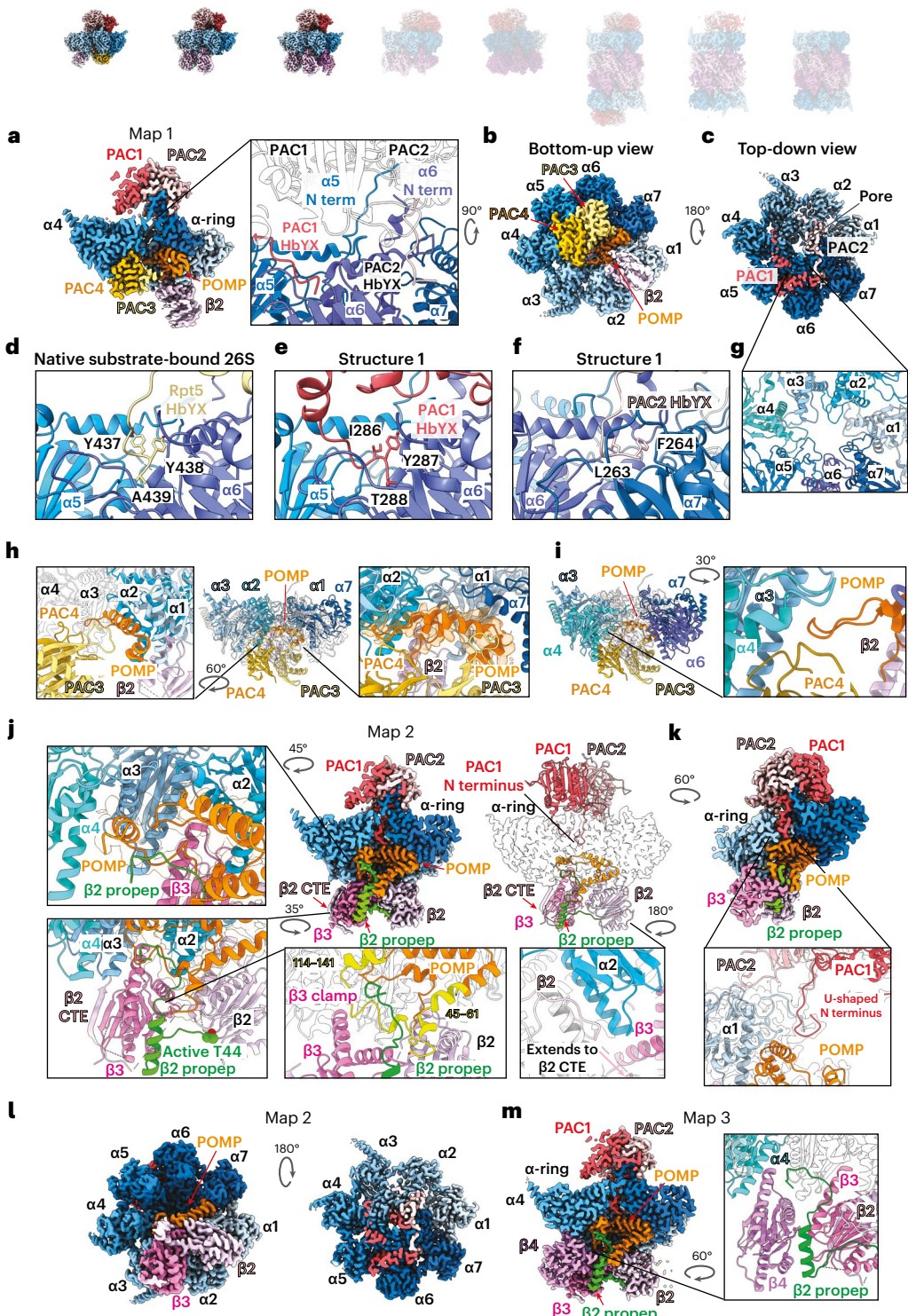

**Fig. 2 | Structural progression of early assembly intermediates.** Cryo-EM maps of eight distinct CP subcomplexes; opaque images represent complexes described in the figure. **a**, Cross-section side view of cryo-EM map of structure 1 showing all five assembly chaperones. PAC1/PAC2 are perched atop the α-ring and interact with all α subunits except for α3. term, terminus. **b**, Bottom-up view of cryo-EM map of structure 1 showing PAC3/PAC4 in the groove between α4–α5 and α5–α6 and β2 in the groove between α1 and α2. **c**, Top-down view of map 1 showing the interaction of PAC1/PAC2 with the α-ring; the central pore is open. **d**, Close-up view of the HbYX motif of Rpt5 inserted between α5 and α6 in native substrate-bound 26S proteasome (PDB 6MSJ). **e**, Close-up view of the HbYX motif of PAC1 inserted between α5 and α6 in structure 1. **f**, Close-up view of the HbYX motif of PAC2 inserted between α6 and α7 in structure 1. **g**, Top-down view of structure 1 (PAC1/PAC2 not shown) showing that the α-ring 'gate' is open.

**h**, Side view of structure 1 showing that PAC3/PAC4 interact with α3–α7 of the α-ring. **i**, Side view of structure 1 showing that POMP interacts with α1–α3 and α7 of the α-ring. **j**, Cross-section view of cryo-EM map of structure 2 showing that the incorporation of β3 is mutually stabilized by the β2 propeptide (propep) and β2 CTE. Close-up view: the additional POMP N-terminal region interacting with the β2 propeptide and β3. The β2 propeptide is colored green. Most of POMP is colored orange, but the designated regions are yellow. **k**, Close-up view of the PAC1 N terminus of structure 2 showing its interaction with α1 and POMP. **l**, Left: bottom-up view of cryo-EM map of structure 2 showing that the release of PAC3/PAC4 liberates grooves between α4–α5 and α5–α6 and that β3 occupies the α2–α3 groove. Right: top-down view showing that the pore is closed. **m**, Cross-section view of cryo-EM map of structure 3 showing the incorporation of β4 and its interaction with β3 and the α-ring.

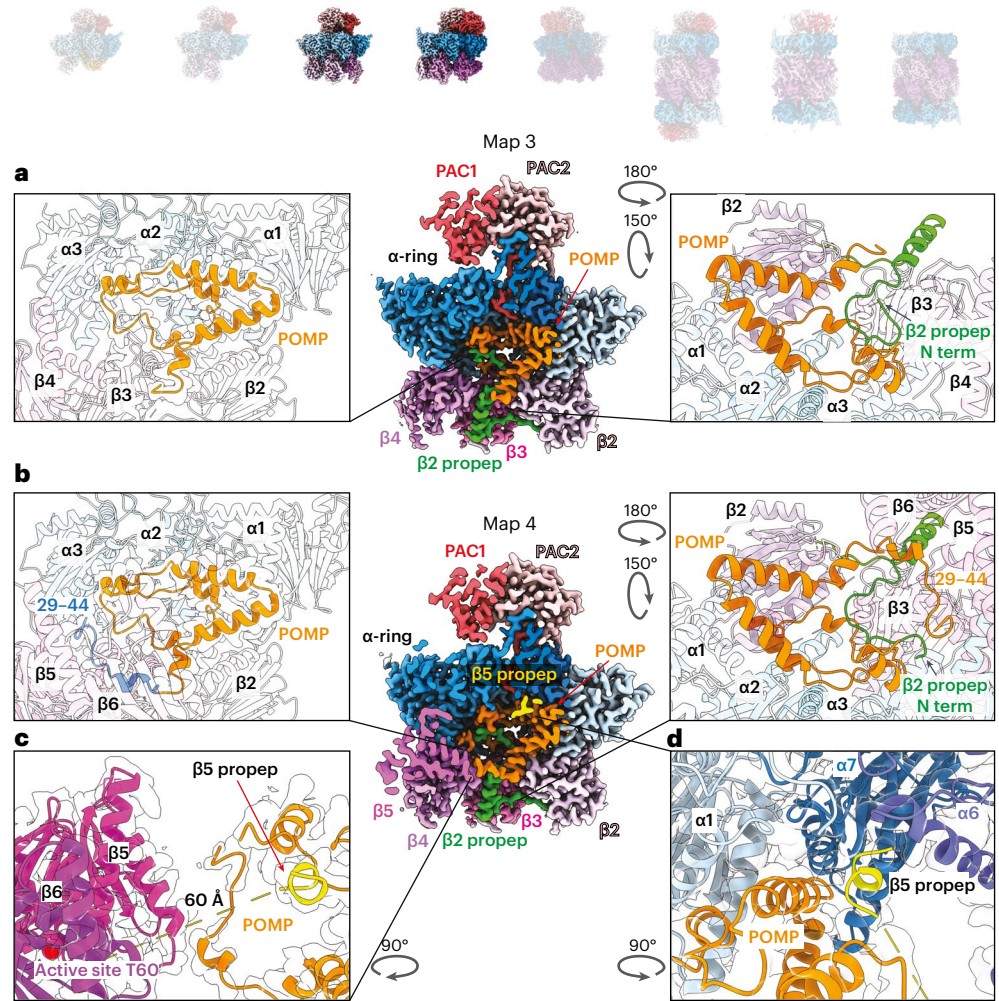

**Fig. 3 | Distinct structural features upon β5 and β6 incorporation.** Cryo-EM maps of eight distinct CP subcomplexes; opaque images represent complexes described in the figure. **a**, Cross-section view of cryo-EM map of structure 3 showing PAC1/PAC2, POMP and β2–β4. Left: close-up view of POMP showing its helix–turn–helix configuration and its interactions with the α-ring and β subunits. Right: close-up view revealing the interaction between POMP and the β2 propeptide. **b**, Cross-section view of cryo-EM map of structure 4 showing PAC1/PAC2, POMP and β2–β5 upon β5 and β6 incorporation (β6 is cropped in the cross-section). Left: close-up view of POMP showing that the additional POMP N-terminal region (blue) is resolved and stabilizes β5 and β6 incorporation. Right: close-up view of the β2 propeptide (green) revealing that the additional resolved POMP N-terminal region interacts with and reorients the N terminus of the β2 propeptide. **c**, Close-up view of β5, β6 and POMP showing that the additional resolved N-terminal region of POMP interacts with the β5 loop. The β5 propeptide is colored yellow. **d**, Close-up view of the β5 propeptide showing its interaction with α6, α7, α1 and POMP.

the subunit center (β4 and β7) (Extended Data Fig. 8b–h). The fusion hairpins and fusion loops become visible and/or are reoriented upon fusion, as seen in the preholo intermediate, where their conformations resemble the mature CP (Extended Data Fig. 8i–k).

Overall, the structural similarity of β1, β2 and β5 between the preholo 20S intermediate and mature 20S CP is consistent, with fusion having a major role in driving autocatalytic activation. When ordered after fusion, the fusion hairpins and fusion loops abut the catalytic Thr residues of β1, β2 and β5, and they shape the active sites of the protease subunits (Fig. 5c–e). Furthermore, the fusion loop contains key residues required for autocatalytic protease activation[53].

For the β2 subunit, there is additional restructuring of the protease domain. In structures 3–5, β2's residues 225–237 face the α-ring and cling to the adjacent α2 subunit (Fig. 5c). This interaction was not observed in proteasome assembly intermediates from yeast because of this α2 region's different sequence and structure (Extended Data Fig. 8l). For human β2, the post-fusion structures show residues 225–237 making a U-turn and proceeding toward the other half-CP (Fig. 5c, middle). This structural remodeling, commensurate with the fusion of

two half-CPs, completes β2's protease fold. This conformational change also directs subsequent β2 residues (237–247) toward the opposite half-CP. These β2 residues were not visible and presumably dynamic in the structures across β-ring assembly, but, after fusion, they formed a belt engaging β6 from the opposite half-CP (Fig. 5c, bottom).

Interestingly, in the preholo 20S CP structure, density corresponding to the propeptides continues from the β2 and β5 active sites (Extended Data Fig. 8m,n). Meanwhile, prior crystallographic and functional analyses of yeast proteasomes harboring variations in the propeptides and residues adjacent to the active site suggested that the rate of propeptide cleavage is influenced by even subtle perturbation of (1) backbone conformation at the junction between the active site and propeptide; (2) adjacent side chains including a critical Lys; and (3) the constellation of surrounding water molecules[40,53]. Although the resolution of our structures precludes precise placement of atoms and visualization of water molecules, the preholo 20S CP intermediate structure shows features poised to impact these facets of propeptide maturation. First, the β2 and β5 propeptides make numerous contacts that impact their orientation. For the β2 propeptide, these

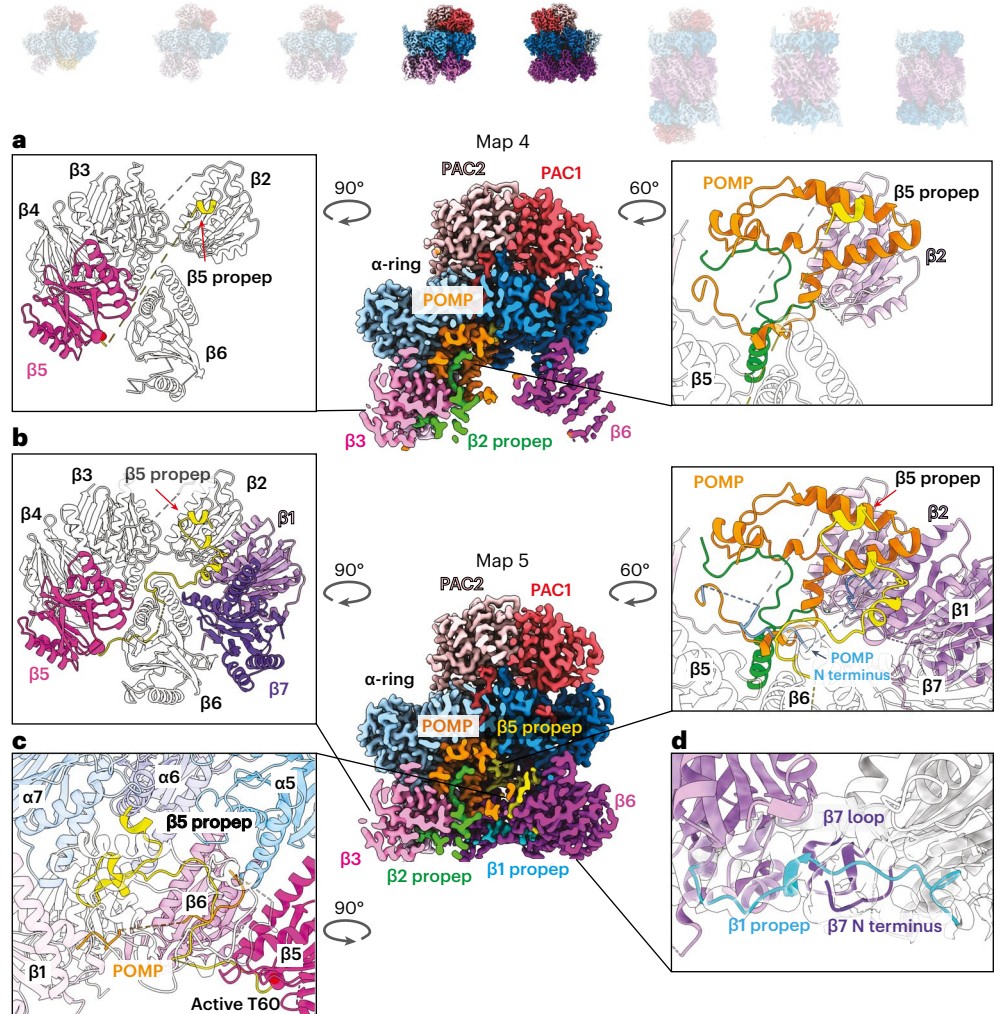

**Fig. 4 | Incorporation of β7 and β1 completes the β-ring and reveals propeptides from all proteolytic subunits.** Cryo-EM maps of eight distinct CP subcomplexes; opaque images represent complexes described in the figure. **a**, Cross-section view of cryo-EM map of structure 4 showing PAC1/PAC2, POMP, β3 and β6 upon β5 and β6 incorporation (β2 is hidden behind, and β4–β5 are cropped in the cross-section). Left: close-up view of β-ring. Right: interaction between β5 propeptide and POMP. The β2 propeptide is colored green, and the β5 propeptide is colored yellow. **b**, Cross-section view of cryo-EM map of structure 5 showing PAC1/PAC2, POMP, β3 and β6 upon β7 and β1 incorporation (β7 and β1–β2 are hidden behind, and β4–β5 are cropped in the cross-section).

Left: close-up view showing that the additional β5 propeptide is resolved upon completion of the β-ring compared to β5 in structure 4. Right: close-up view of the β5 propeptide's interaction with POMP, β1 and β2 showing that POMP's N terminus (blue) is resolved upon completion of the β-ring and interacts with the β5 propeptide. The β2 propeptide is colored green, and the β5 propeptide is colored yellow. **c**, Close-up view of the β5 propeptide's interaction with α5–α7, β5–β6 and β1. **d**, Close-up view of newly incorporated β1 and β7 revealing the β1 propeptide and its interaction with the β7 N terminus and loop. The β1 propeptide is colored cyan.

mirror those in structure 5 and include interactions with β3, β4, α3 and POMP (Extended Data Fig. 8m). Meanwhile, residues 52–59 of the β5 propeptide retain contacts with β6 (Fig. 5d). Second, the propeptides align their active site and adjacent residues critical for their cleavage[53] (Fig. 5c–e).

Although the human preholo 20S CP exhibits two-fold symmetry and maintains much of the assembly factor scaffold, only a single PAC1/PAC2 is visible in the premature complex (Fig. 5a). Notably, this premature configuration was proposed in the original description of PAC1/PAC2-mediated assembly[35] and observed, albeit at >20 Å resolution, for yeast[63]. We presume that an intermediate with double capping of the 20S CP by PAC1/PAC2 and with POMP already released exists but has not been observed. We speculate that the combination of a relatively lower abundance than for the preholo intermediate and a larger size than for the mature 20S CP limited the number of particles in the chromatography fraction we examined by cryo-EM. Nonetheless, the premature

20S CP structure suggests that PAC1/PAC2 elimination follows POMP degradation. Although the ~5 Å local resolution at the two ends of the premature 20S CP precludes definitive mechanistic conclusions, we note that the PAC1/PAC2-bound side superimposes well with the corresponding portion of structure 1 (Supplementary Video 10) and with a crystal structure of yeast Pba1/Pba2-bound mature CP[49].

The preholo, premature and mature 20S CP complexes also show differences in the α-ring, which primarily impact gating and accessibility of the central pore (Supplementary Video 10). The α-ring is similar across structures 2–5 and the preholo 20S CP structure (Supplementary Video 7), where it is largely occluded by elements coming from both directions (Extended Data Fig. 5g). The α1 and PAC1 N termini point from the outside toward the inside of the barrel. From the inside, POMP binds PAC1's N terminus to further occupy the pore. These elements are not visible in the premature 20S CP structure, but the majority of the N termini from α subunits are still directed outward due to engagement

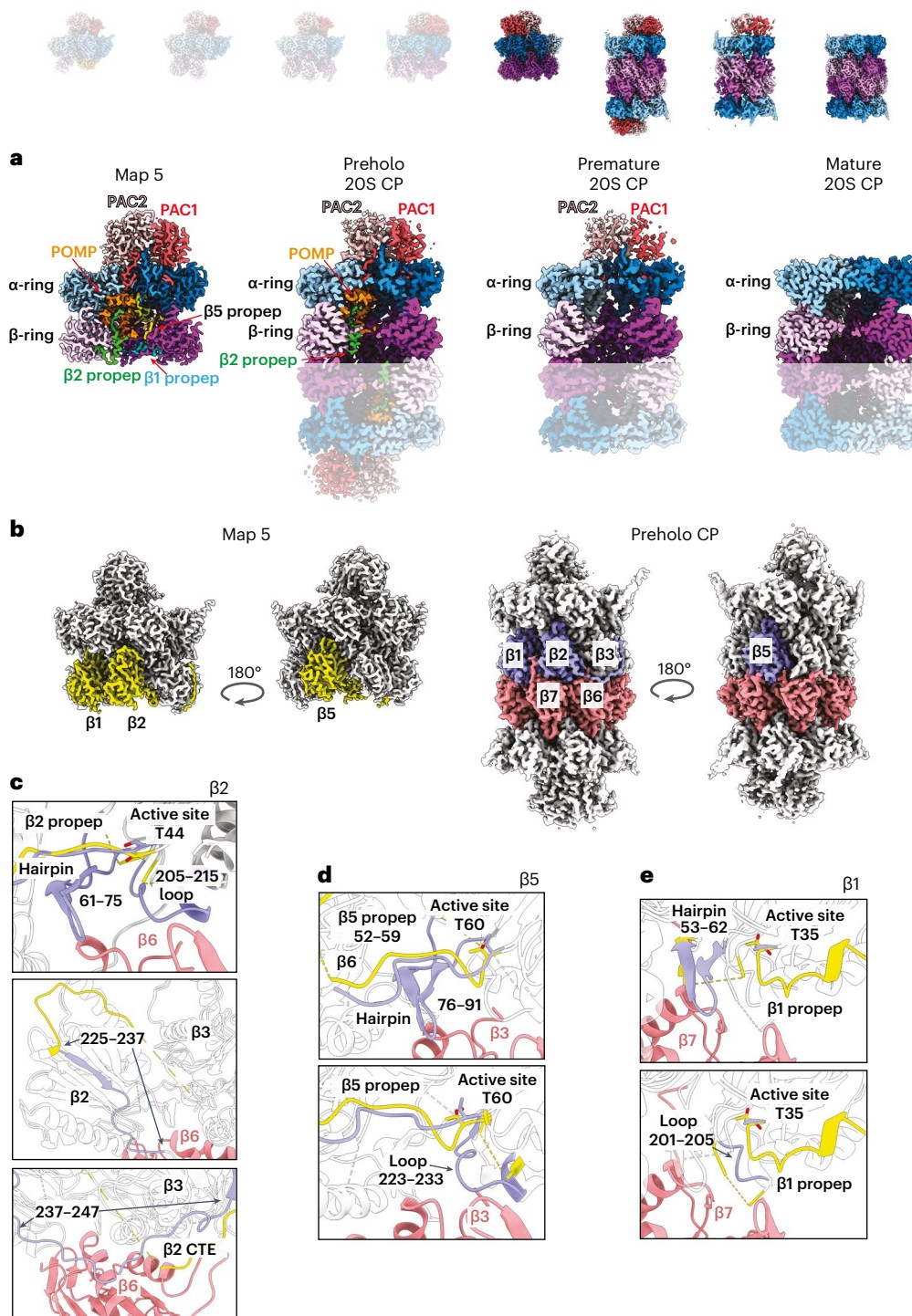

**Fig. 5 | Half-CP fusion and 20S CP maturation.** Cryo-EM maps of eight distinct CP subcomplexes; opaque images represent complexes described in the figure. **a**, Cross-section view of cryo-EM map of structure 5, preholo 20S CP, premature 20S CP and mature 20S CP showing the earlier presence and subsequent absence of assembly chaperones and β propeptides during the 20S CP fusion and maturation process. **b**, Cryo-EM map of structure 5 with β1, β2 and β5 colored in yellow. Cryo-EM map of preholo 20S CP with β1, β2 and β5 of one half-CP colored in purple and the other half-CP colored in salmon. **c**, Close-up view of β2's structural changes (β2 from structure 5 colored in yellow and β2 from preholo 20S CP colored in purple) and interactions with β6 (salmon) from the opposite half-CP upon CP fusion. The active site position is shown and fusion hairpin and fusion loop elements are labeled 'hairpin' and 'loop', respectively. **d**, Close-up view of β5's structural changes (β5 from structure 5 colored in yellow and β5 from preholo 20S CP colored in purple) and interactions with β3 (salmon) from the opposite half-CP upon CP fusion. The active site position is shown and fusion hairpin and fusion loop elements are labeled 'hairpin' and 'loop', respectively. **e**, Close-up view of β1's structural changes (β1 from structure 5 colored in yellow and β1 from preholo 20S CP colored in purple) and interactions with β7 (salmon) from the opposite half-CP upon CP fusion. The active site position is shown and fusion hairpin and fusion loop elements are labeled 'hairpin' and 'loop', respectively.

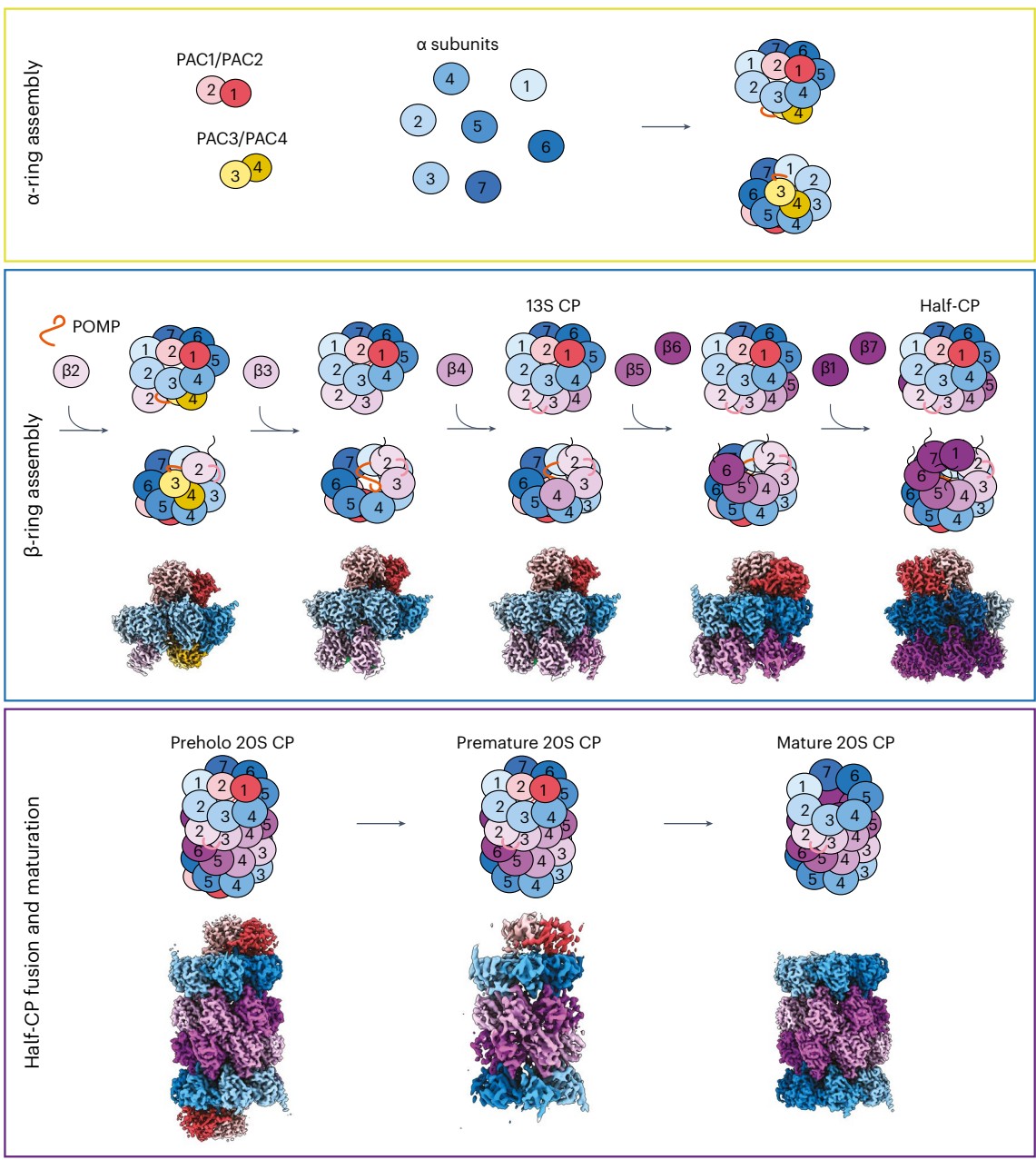

**Fig. 6 | Schematic representation of chaperone-mediated stepwise assembly of the 20S CP and summary of high-resolution structures of human assembly intermediates.** Top: assembly of the 20S CP through α-ring assembly (yellow outline). Middle: β-ring assembly (blue outline). Bottom: half-CP fusion and maturation (purple outline). High-resolution maps of human CP assembly intermediates presented in this study are indicated below their schematic representations.

of PAC1/PAC2. PAC1/PAC2, thus, enforce the open-gate conformation until the very end of CP maturation. The premature 20S CP would set the stage for the N termini of α2–α5 to fill the pore from the opposite side of PAC1/PAC2's anchor, thereby adopting the closed-gate conformation of the mature 20S CP (Supplementary Video 10).

## Discussion

Given their central cellular roles, it is important to understand how proteasomes are generated (Fig. 6). The seven cryo-EM reconstructions of recombinant human CP assembly complexes reported here include structures not previously observed for several proteasome assembly intermediates, including an intermediate containing PAC3/PAC4. Structure 1 reveals how the β-ring is initiated and why β2 is the first subunit to incorporate. This structure also reveals two probable

and unexpected contributions of PAC3/PAC4 to the β-ring. First, PAC3/PAC4 directly bind POMP. Second, PAC3/PAC4 may help to order the incorporation of β subunits by preventing the untimely incorporation of later subunits through occlusion of their binding sites. Structure 2, when compared to structures 1 and 3, shows progressive steps by which individual subunits are added. Structure 5, the half-CP, reveals the function of the β5 propeptide in completing β-ring assembly and its collaboration with POMP in this process; structure 5 also shows how the proteasome active sites are restrained before half-CP fusion.

Structures 3 and 4, as well as the human preholo intermediate, also allow comparison to previously determined structures of the corresponding yeast 13S, pre-15S and preholo intermediates[56,64], both validating our approach and revealing conserved or species-specific aspects of assembly. A striking feature was the overall high degree

of structural conservation of POMP and the β2 and β5 propeptides, despite relatively low sequence similarity between species.

The structures also reveal key conceptual principles concerning the directionality of CP biogenesis, which may have relevance for other molecular machines. In principle, the assembly of any 28-subunit complex could proceed through an almost infinite number of distinct pathways. In mature CP, each subunit is engaged in lateral, vertical and even diagonal interactions with its neighbors. These interactions are reflected in the exceptional stability of the CP, which can withstand harsh purification conditions (for example, 0.5 M NaCl)[56,71]. These interfaces cannot be fully occupied during CP assembly. Many unpartnered regions of proteasome subunits and assembly factors appear flexible, as reflected in the lack of density in the cryo-EM maps. Each incoming subunit 'structures' or rigidifies its neighboring subunit or chaperone in a templating-like manner. Because each interface is unique, binding affinity should naturally be highest for true neighbors, which should drive the pathway forward through energetic favorability. Insertion of each new subunit not only stabilizes the existing assembly intermediate, but also promotes the formation of the next intermediate until the β-ring is complete. By reducing the number of available assembly pathways, this should increase the fidelity of biogenesis. Here, this ensures sequestration of the proteasome's destructive capabilities inside the gated CP.

Untimely incorporation of subunits may also be prevented by a series of previously unknown structural checkpoints. For example, the chaperones initiating β-ring assembly—PAC3/PAC4 and POMP— together bind all α subunits, ensuring that β-ring assembly is initiated only after completion of the α-ring. As another example, several structural features require proper incorporation of β2–β4 before β5. In turn, the β6, β1 and β7 subunits, which are secured by the β5 propeptide, are stabilized only after β5 insertion.

We demonstrate roles for the active site subunit propeptides in helping to recruit and stabilize their neighboring subunits, which are achieved not in isolation but through a complex network of interactions with POMP. Indeed, one can conceive of CP assembly as a process of progressively building an internal structured scaffold within the growing complex (Supplementary Video 8) and then self-disassembling that scaffold once proteolytic activation has occurred (Supplementary Video 9).

Interestingly, higher organisms encode additional β subunits that are incorporated into immune-specific proteasomes[28]. Differences between the canonical sequences used here and the immunoproteasome mapped on structures 1–5 show sequence variations in propeptides and several other residues making contacts along the assembly process but that are exposed in the mature complex (Extended Data Fig. 9a,b). Thus, future studies may reveal distinct assembly features of specialized proteasomes.

Similar to β-ring assembly, but on a larger scale, complementary protein segments not observed in the chaperone-bound half-CP (structure 5) simultaneously become visible in the preholo intermediate. These include what we refer to as fusion hairpins and fusion loops, which contribute to the active site configuration in β1, β2 and β5 and have been implicated in autocatalytic propeptide processing[53]. Interestingly, the one subunit that shows little change between the half-CP and preholo CP is β6. In contrast, its midline partner, β2, shows the most large-scale remodeling during half-CP fusion. β6 may assist in templating this process, with its preformed fusion hairpin and fusion loop capturing those from the opposing β2. It is possible that β6 ensnaring the opposite β2 CTE loop acts as a lever that redirects the preceding sequence and promotes completion of the adjacent protease fold.

Meanwhile, structural progression from preholo to premature to mature 20S CP illuminates the final stages of assembly. As neither POMP nor the propeptides are visible in premature 20S, propeptide cleavage might also destabilize the internal assembly factor scaffold, allowing for its degradation. PAC1/PAC2 binding, in turn, might be destabilized

by loss of contacts between PAC1's N terminus and POMP. Future studies will be required to determine whether PAC1/PAC2 are degraded during assembly or in isolation after liberation from mature CP.

Defects in CP assembly are responsible for a growing family of related diseases characterized by immune dysregulation[27,43–45]. Our study provides a system to determine the functional consequences of patient-derived variations for CP assembly. In the long term, a more complete understanding of CP assembly in normal and diseased states might lead to therapeutic interventions to enhance or impair the efficiency of CP biogenesis. Lastly, the strategy used here to visualize sequential intermediates along complex assembly pathways could be applied to other molecular machines, which in turn could be helpful in understanding diseases characterized by defects in multiprotein complex biogenesis.

## Online content

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

## Methods

### Strategy to visualize human 20S proteasome assembly intermediates

**Considerations.** To date, structural knowledge of chaperone-bound eukaryotic 20S CP assembly intermediates has largely relied on strategies applied to budding yeast[47,49,56,63,64], including (1) the use of mutant strains that stall the assembly and, thus, increase the cellular abundance of such intermediates; (2) affinity tagging a chaperone (Ump1) to purify its associated proteins; (3) mixing separately purified chaperone and CP complexes; and (4) coexpression and purification of subcomplexes for X-ray crystallography. Meanwhile, it would seem that the roughly tenfold-lower cellular concentrations of assembly chaperones compared to α subunits and β subunits[72], as well as the purification procedures to enrich for fully assembled complexes, would pose challenges for structurally characterizing the intermediates along human 20S CP assembly. However, ever since earlier reports of PAC1/PAC2 and PAC3, there has been evidence of intermediate chaperone assemblies in mammalian cells[35,36]. More recent proteome-wide interactome studies performed by immunoprecipitation with mass spectrometry of C-terminally tagged β subunits indicated interaction with not only other stable proteasome components, but also the chaperones PAC1–PAC4 (PSMG1–PSMG4) and POMP. Endogenously tagged 20S CP components were shown to interact not only with proteasome subunits, but also with chaperones at substantially lower stoichiometry[72]. Moreover, purifications based on exogenously expressed affinity-tagged β2 (PSMB7)−but not β7 (PSMB4)−showed the early-departing assembly factors PAC3 and PAC4 (PSMG3 and PSMG4). Thus, it stands to reason that the challenges when enriching human 20S CP intermediates could in principle be overcome by (1) increasing the expression of assembly chaperones; (2) affinity purification based on tagging the first β subunit (β2) to be incorporated; and (3) size fractionation such that complexes within a sample would be of relatively more similar mass. To achieve this, we used systems for high-level expression of protein complexes in insect cells.

**Cloning.** The complementary DNA (cDNA) encoding 20S CP subunits and assembly chaperones was either obtained from an in-house human cDNA library (Max Planck Institute of Biochemistry) or ordered as synthetic cDNA from Eurofins or IDT, if multiple restriction sites were to be occluded. Baculovirus transfer vectors were assembled using a combination of the biGBac system[61] and third-generation MultiBac system[62]. All primers used are listed in Supplementary Table 1. In short, all subunits were first cloned by Gibson assembly into the vector pACEBac1, which was used as a library vector. For β2 (*PSMB7*) and β7 (*PSMB4*), both untagged and tobacco etch virus (TEV)-cleavable C-terminal twin-Strep tag constructs were generated. Primers for step 1 assembly of the biGBac system were modified to prime before the *polh* promoter and SV40pA site within pACEBac1 (ref. 62). Step 1 assembly reactions were set up to obtain the vectors pACEBac1-POMP-PSMG1-PSMG2-PSMG3-PSMG4 (to express POMP, PAC1, PAC2, PAC3 and PAC4; Addgene plasmid 214137, Research Resource Identifier (RRID) Addgene_214137), pACEBac1-PSMA1-PSMA2-PSMA3-PSMA4 (Addgene plasmid 217893, RRID Addgene_217893), pACEBac1-PSMA5-PSMA6-PSMA7 (Addgene plasmid 217894, RRID Addgene_217894), pACE-Bac1-PSMB1-PSMB2-PSMB3-PSMB4 (Addgene plasmid 217895, RRID Addgene_217895), pACEBac1-PSMB5-PSMB6-PSMB7-TEV-2xSTII (Addgene plasmid 217896, RRID Addgene_217893217896), pACEBac1-PSMB1-PSMB2-PSMB3-PSMB4-TEV-2xSTII (Addgene plasmid 217897, RRID Addgene_217897) and pACEBac1-PSMB5-PSMB6-PSMB7 (Addgene plasmid 217898, RRID Addgene_217898). Subsequently, the final vectors pACEBac1-PSMA1-PSMA2-PSMA3-PSMA4-PSMA5-PSMA6-PSMA7 (to express α6, α2, α7, α3, α5, α1 and α4; Addgene plasmid 214138, RRID Addgene_214138), pACEBac1-PSMB1-PSMB2-PSMB3-PSMB4-PSMB5-PSMB6-PSMB7-TEV-2xSTII (to express β6, β4, β3, β7, β5, β1 and β2 with C-terminal twin-Strep tag; Addgene plasmid 214139, RRID

Addgene_214139) and pACEBac1-PSMB1-PSMB2-PSMB3-PSMB4-TEV-2xSTII-PSMB5-PSMB6-PSMB7 (to express β6, β4, β3, β7 with C-terminal twin-Strep tag, β5, β1 and β2; Addgene plasmid 214140, RRID Addgene_214140) were assembled by restriction cloning using the MultiBac multiplication module.

**Protein expression and purification.** *Sf*9 insect cells (Thermo Fisher Scientific, catalogue number 11496015, RRID CVCL_0549) were cultured in serum-free EX-CELL 420 medium (Sigma-Aldrich, catalogue number 24420C), and *Trichoplusnia ni* High Five cells (Thermo Fisher Scientific, catalogue number B85502, RRID CVCL_C190) were cultured in protein-free ESF 921 insect cell culture medium (Expression Systems, catalogue number NC9036111). Bacmid preparations, virus amplification in *Sf*9 cells and protein expression in *T. ni* High Five insect cells were conducted according to standard protocols[73,74].

Mature 20S CPs, together with their assembly intermediates, were purified by standard affinity purification on StrepTactin Sepharose High Performance (Cytiva, catalogue number GE28-9355-99) and subsequent SEC on a Superose6 Increase 10/300 (Cytiva, catalogue number GE29-0915-96) column in 25 mM HEPES pH 7.5 (KOH), 150 mM NaCl and 1 mM dithiothreitol.

### Structure determination by transmission cryo-EM

**Sample preparation.** Cryo-EM grids were prepared for three samples. From the β2-tagged and β7-tagged affinity purifications, the peak fractions (C2) were diluted to a concentration between 0.5 and 0.7 mg ml⁻¹. From the β2-tagged affinity purification, SEC fractions containing 20S CP assembly intermediates (C4−C7) were pooled and concentrated to 5 mg ml⁻¹; then, Fos-Choline-8 was added to a final concentration of 0.25 mM. Quantifoil R1.2/1.3 Cu-200 grids (Quantifoil Micro Tools) were glow-discharged for 30 s in a PDC-32G-2 (Harrick Plasma); 3.5 µl samples were applied and plunge-frozen in a liquid ethane−propane mix with a Vitribot Mark IV (Thermo Fisher Scientific) at 4 °C and 100% humidity.

**Data acquisition and processing.** For cryo-EM data acquisition, all grids were prescreened for optimal particle distribution and ice thickness on a Glacios cryo-transmission EM (cryo-TEM) instrument (Thermo Fisher Scientific) operated at 200 kV, equipped with a K2 Summit direct electron detector (DED) camera (Gatan). Data collection for the samples from the peak fractions was carried out using a Glacios cryo-TEM microscope. Data collection was set up with SerialEM (RRID SCR_017293, http://bio3d.colorado.edu/SerialEM/) version 4.1 (ref. 75) using coma-corrected beam-image shift. One movie per hole was recorded in counting mode with a 3 × 3 multihole record acquisition scheme at a pixel size of 1.181 Å per pixel with a nominal magnification of ×36,000 for the β2-tagged and β7-tagged peak fraction datasets or at a pixel size of 1.885 Å per pixel with a nominal magnification of ×22,000 for the prehole and premature 20S CP datasets. A total dose of 56−60 e⁻ per Å² was fractionated over 40 frames, with a target defocus range of −1.0 µm to −2.6 µm.

The dataset yielding the 20S CP assembly intermediate structures was collected on a Titan Krios G2 cryo-TEM (Thermo Fisher Scientific) operated at 300 kV equipped with a Bio Quantum post-column energy filter (Gatan; 10 eV) and K3 DED camera (Gatan). Data collection was set up with SerialEM version 4.1 using coma-corrected beam-image shift. Three movies per hole were recorded with a 5 × 5 multihole record acquisition scheme at a pixel size of 0.8512 Å per pixel with a nominal magnification of ×105,000 in counting mode (no correlated double sampling). A total dose of 67 e⁻ per Å² was fractionated over 30 frames, with a target defocus range of −1.0 µm to −2.6 µm.

Data processing was performed as follows. The Glacios K2 dataset was processed with cryoSPARC version 4.2 (ref. 76) (RRID SCR_016501; for details, see Extended Data Fig. 1). The raw movies of the Titan Krios K3 dataset were motion-corrected on the fly with Focus[77]

(RRID SCR_014462), and motion-corrected micrographs were imported into cryoSPARC for all subsequent processing steps (including patch contrast transfer function estimation, blob picking, template picking, two-dimensional classification, heterogenous refinement, nonuniform refinement, manual sharpening, local resolution estimation, local filtering and final postprocessing with DeepEMhancer[78] (https://github.com/rsanchezgarc/deepEMhancer); for details, see Extended Data Fig. 2).

Importantly, the cryo-EM data showed that the peak fractions from both the complex based on the β2 tag and the complex based on the β7 tag were dominated by fully assembled, mature 20S CPs (Extended Data Figs. 1 and 2). In accordance with the expression and purification strategy, map 5 (termed 5[7] for the β7-tagged complex) was refined for both purifications. Map 5 for the β2-tagged complex was superior, presumably because of the purification enriching for complexes in this molecular weight range, as well as the quality and quantity of data collected on the Titan Krios. Meanwhile, the refinement of maps 1–4 correlates with the purification strategy based on the β2 tag and enrichment of complexes in their molecular weight range.

**Model building and refinement.** Model building and refinement were performed as follows. AlphaFold2 (refs. 79,80) (RRID SCR_023662, http://github.com/google-deepmind/alphafold) models of PAC1–PAC4 and POMP, along with corresponding chains from a published model of the 20S CP (PDB 5LE5), were manually docked with ChimeraX version 1.5 (ref. 81) (RRID SCR_015872) in all maps. Atomic models were locally adjusted in Coot (RRID SCR_014222, http://www2.mrc-lmb.cam.ac.uk/personal/pemsley/coot/) version 0.9.8.7 (ref. 82) and iteratively refined with Phenix version 1.19.2 (ref. 83) (RRID SCR_014224). We modeled into density when the trajectory of the backbone was unambiguous, and side chains were placed when density was observed in manually sharpened or automatically sharpened maps. There are additional regions of density around β2 in structure 1, β3 in structure 2, β4 in structure 3, β5 in structure 4 and β1 or β7 in structure 5, for which coordinates were not modeled because of the ambiguity of these regions. The unmodeled density presumably corresponds to regions of these subunits and/or their neighbors that become ordered upon additional interactions formed during assembly. For structures 1–5, the density for the β2 CTE strand that grasps β3 partially overlaps with this element in the mature CP. However, due to a different trajectory for the preceding residues, we could not independently unambiguously assign the sequence. Thus, this was modeled on the basis of the sequence in the mature 20S CP. Final refinement was performed using ISOLDE[84] (https://isolde.cimr.cam.ac.uk/) followed by Phenix refinement. Sequence alignments were performed with Clustal Omega[85] (RRID SCR_001591). For visualization, figures were generated with UCSF ChimeraX and Adobe Illustrator 2023 (RRID SCR_010279). Details of data acquisition, data collection and structure refinement are provided in Table 1.

**Reporting summary**

Further information on research design is available in the Nature Portfolio Reporting Summary linked to this article.

## Data availability

Cryo-EM maps were deposited to the EMDB with the following accession codes: for β2 affinity purification, map 1: EMD-18755, map 2: EMD-18757, map 3: EMD-18758, map 4: EMD-18773, map 5: EMD-18759, preholo 20S CP: EMD-18761, premature 20S CP: EMD-18762 and 20S CP: EMD-18760; for β7 affinity purification, map 5: EMD-19342 and 20S CP: EMD-19343. Atomic coordinates were deposited to the Research Collaboratory for Structural Bioinformatics (RCSB) PDB with the following accession codes: structure 1, PDB 8QYJ; structure 2, PDB 8QYL; structure 3, PDB 8QYM; structure 4, PDB 8QZ9; structure 5, PDB 8QYN; preholo 20S CP, PDB 8QYS; 20S CP, PDB 8QYO. Baculovirus transfer vectors for recombinant production of 20S CPs were submitted

to Addgene (214137, 214138, 214139, 214140, 217893, 217894, 217895, 217896, 217897 and 217898). Detailed protocols associated with this work are available on protocols.io (https://doi.org/10.17504/protocols.io.e6nvwdrozlmk/v1) (ref. 86).

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

## Acknowledgements

We thank D. Bollschweiler and T. Schäfer for the MPIB cryo-EM facility and assistance with data collection. We are grateful to J. Rajan Prabu for maintaining processing infrastructure and assisting with structural model building. This work was supported by Aligning Science Across Parkinson's (ASAP, grant 000282), the Max Planck Society and the Leibniz Prize from Deutsche Forschungsgemeinschaft (DFG, German Research Foundation)—SCHU 3196/1-1 (to B.A.S.), as well as the National Institutes of Health (NIH) grants R01-GM144367 (to J.H.), R01-AG011085 and R01-NS083524 (to J.W.H.). The Michael J. Fox Foundation administers the grant ASAP-000282 on behalf of ASAP and itself. E.A.G. was the recipient of a postdoctoral fellowship from the Edward R. and Anne G. Lefler Center for the Study of Neurodegenerative Disorders. The funders had no role in study design, data collection and analysis, decision to publish or preparation of the manuscript. For the purpose of open access, the authors have applied a CC-BY public copyright license to the author-accepted manuscript version arising from this submission.

## Author contributions

F.A. and B.A.S. designed the research; F.A. contributed new reagents; F.A. and J.D. performed the research; F.A., J.D., S.v.G., S.R., R.M.W.,

E.A.G. and J.W.H. analyzed the data; J.W.H., J.H. and B.A.S. acquired funding; F.A., J.D., E.A.G., J.H. and B.A.S. wrote the paper with input from all authors.

## Funding

## Competing interests

J.W.H. is a founder and consultant for Caraway Therapeutics. B.A.S. is on the scientific advisory boards of Biotheryx and Proxygen. The other authors declare no competing interests.

## Additional information

**Extended data** is available for this paper at https://doi.org/10.1038/s41594-024-01268-9.

**Correspondence and requests for materials** should be addressed to Frank Adolf, John Hanna or Brenda A. Schulman.

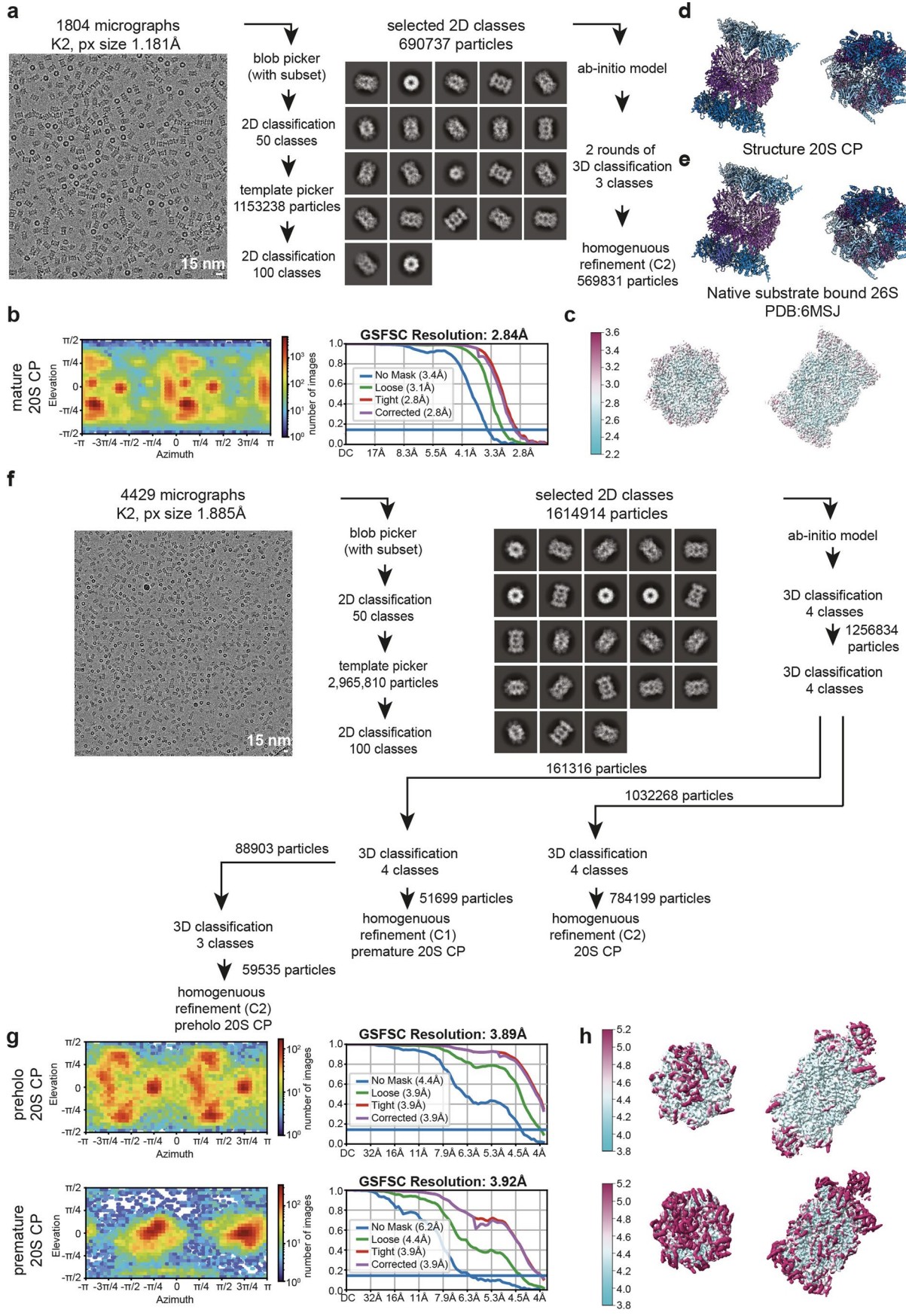

**Extended Data Fig. 1 | See next page for caption.**

**Extended Data Fig. 1 | Structure determination of recombinant β2-tagged human 20S. proteasome. a**, Cryo-EM image processing scheme for β2-tagged mature human 20S proteasome. **b**, Angular distribution and GSFSC curves of human 20S proteasome map. **c**, Local resolution of the cryo-EM map of human 20S proteasome map. **d**, Structure of recombinant human 20S CP with tag on β2 C terminus. **e**, Structure of substrate-bound native human 26S proteasome (PDB: 6MSJ, only 20S core particle is shown). **f**, Cryo-EM image processing scheme for preholo 20S CP and premature 20S CP. **g**, Angular distribution and GSFSC curves of preholo 20S CP (top) and premature 20S CP (bottom) maps **h**, Local resolution of the cryo-EM maps of preholo 20S CP (top) and premature 20S CP (bottom). Scale bar in all micrographs represents 15 nm.

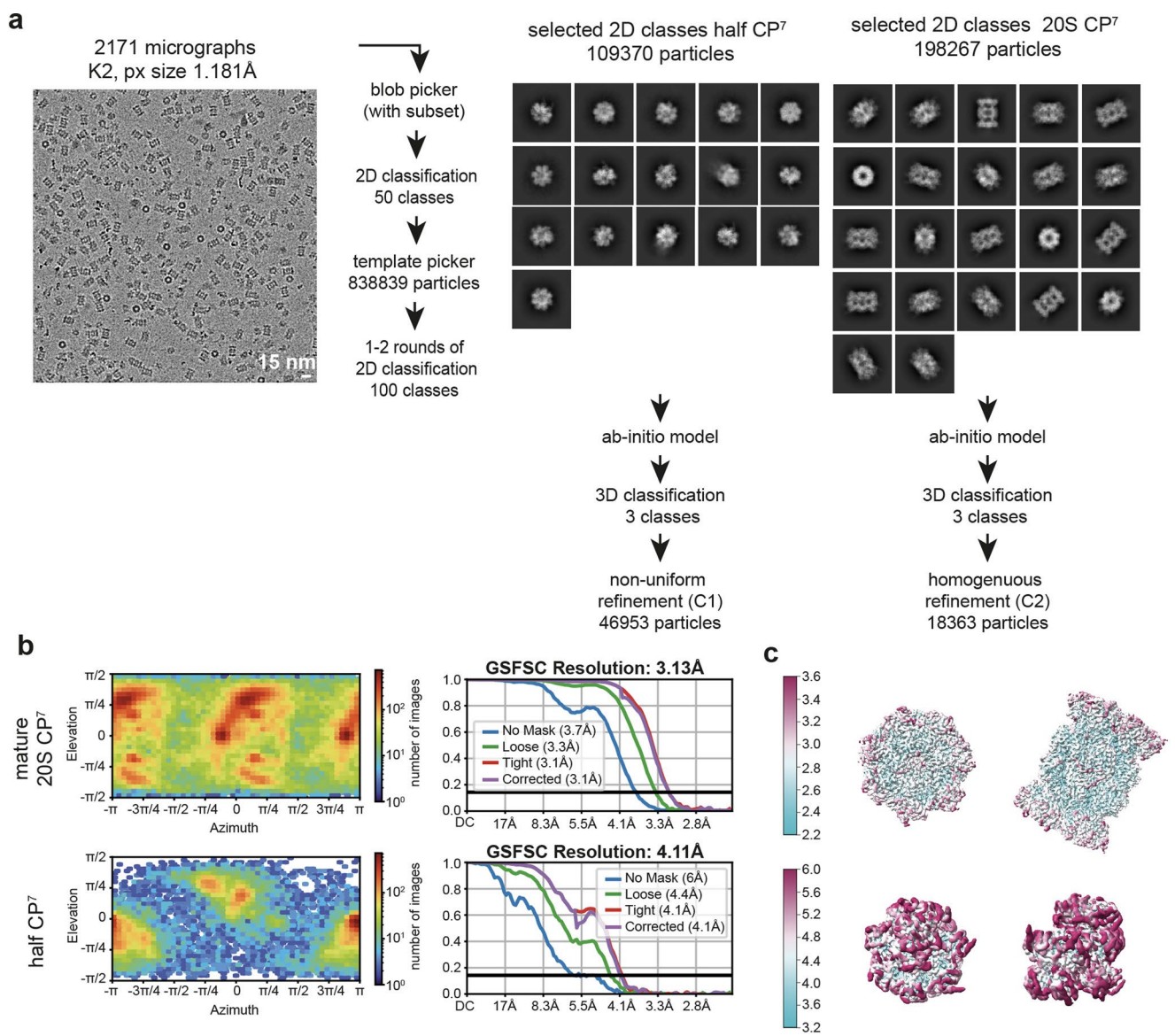

**Extended Data Fig. 2 | Structure determination of recombinant human 20S proteasome and half proteasome with tag on β7. a**, Cryo-EM image processing scheme for human 20S CP and half CP with tag on β7 (indicated by "7" in superscript). **b**, Angular distribution and GSFSC curves of human 20S CP and half CP with tag on β7. **c**, Local resolution of the cryo-EM maps of human 20S CP and half CP with tag on β7. Scale bar in all micrographs represents 15 nm.

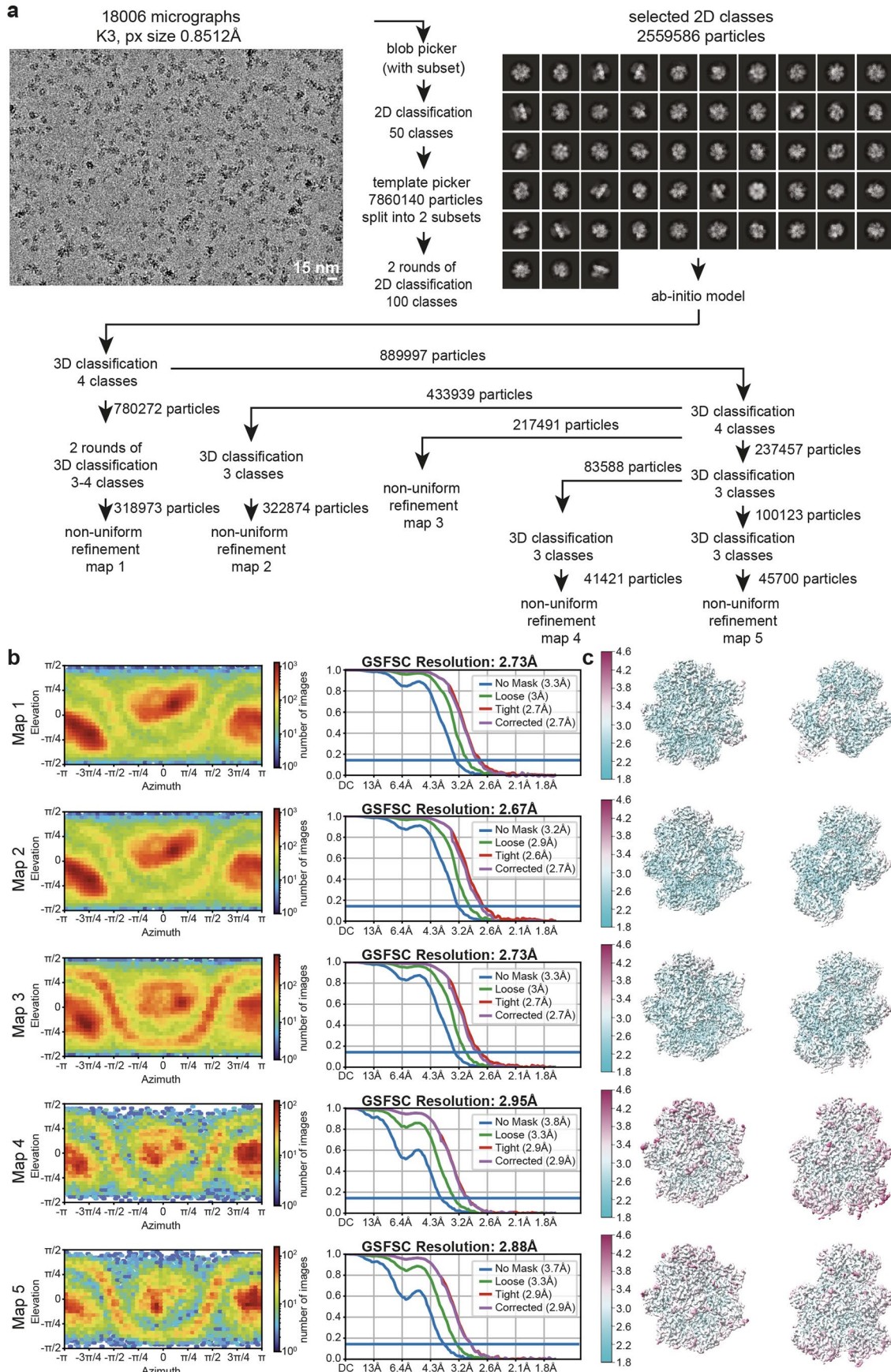

**Extended Data Fig. 3 | Structure determination of recombinant human proteasome assembly intermediates. a**, Cryo-EM image processing scheme for the human proteasome assembly intermediates. **b**, Angular distribution and GSFSC curves of human proteasome assembly intermediates. **c**, Local resolution of the cryo-EM maps of human proteasome assembly intermediates. Scale bar in all micrographs represents 15 nm.

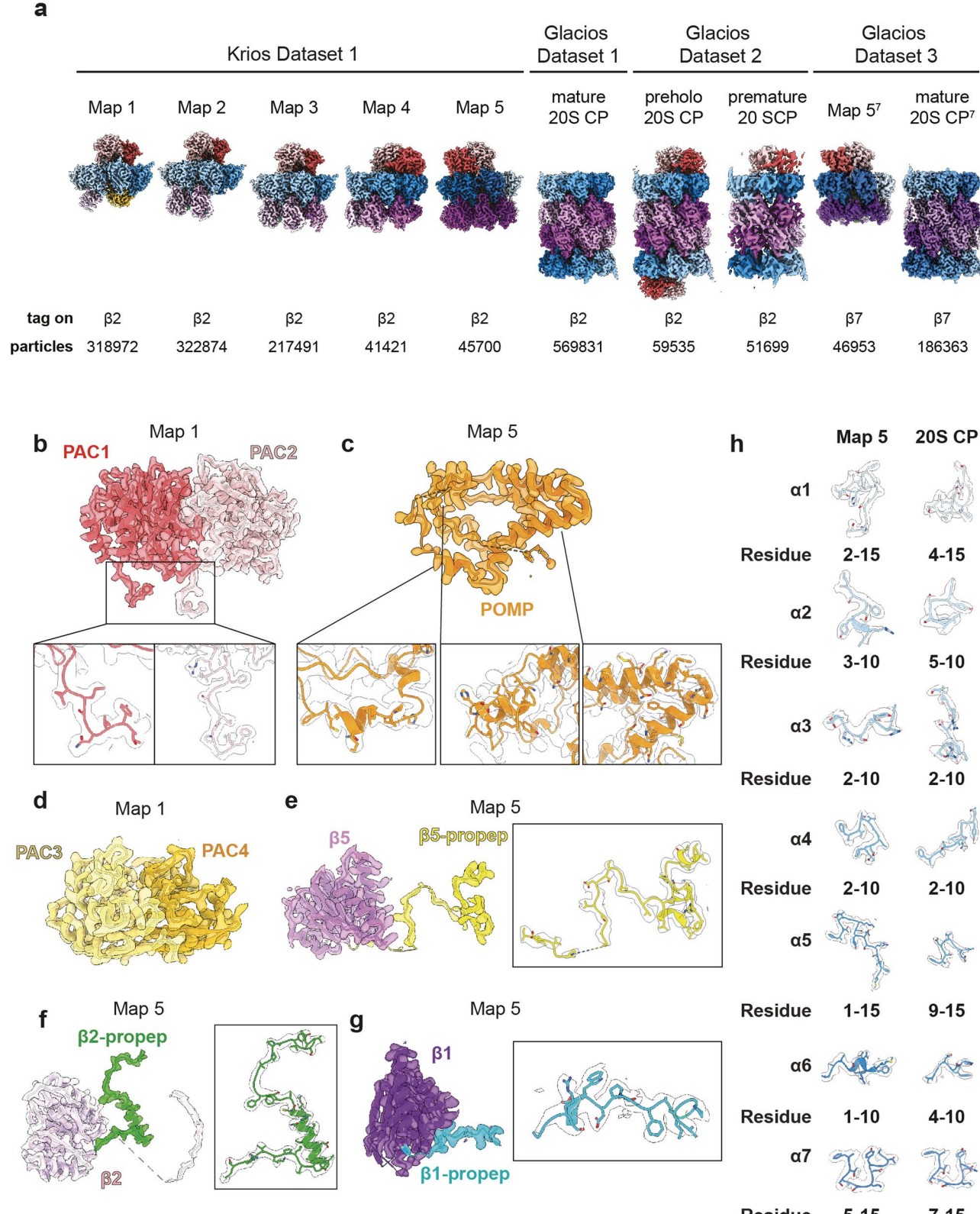

**Extended Data Fig. 4 | Particle proportion of 20S assembly intermediates and 20S CP and EM density of assembly chaperones and β propeptides. a**, Particle distribution of 20S assembly intermediates and 20S CP obtained in 4 datasets. **b**, Model-fitted EM densities of PAC1/2 HbYX motifs. **c**, Model-fitted EM densities of POMP. **d**, Model-fitted EM densities of PAC3/4. **e**, Model-fitted EM densities of β5-propeptide. **f**, Model-fitted EM densities of β2-propeptide **g**, Model-fitted EM densities of β1-propeptide. **h**, Model-fitted EM densities of α subunit N-termini in Map5 and 20S CP. Each pair is globally aligned to show conformational changes at N-termini.

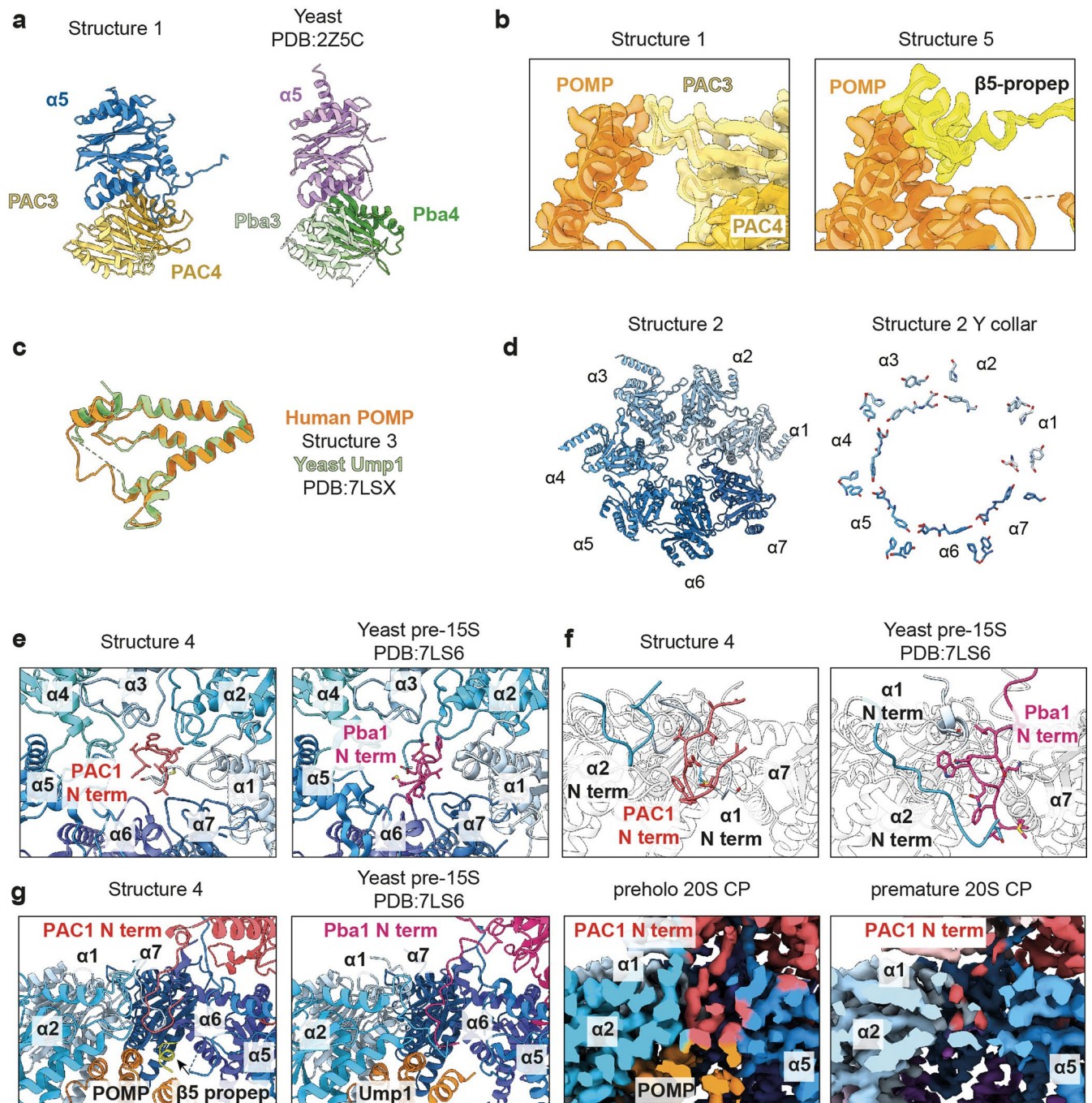

**Extended Data Fig. 5 | Structural comparison of human 20S CP assembly intermediates with yeast orthologs and 20S CP with 26S proteasome. a**, Comparison of human PAC3/4 (yellow) with yeast Pba3/4 (green, PDB:2Z5C) reveals their structural similarity. **b**, Comparison of Structure 1 and Structure 5 shows the position of PAC3/4 in Structure 1 is occupied by β4-7 later in Structure 5. **c**, Comparison of human POMP (orange) with yeast Ump1 (green, PDB:7LSX) reveals their structural similarity. **d**, Y collar of α-ring in Structure 1 and 2 shows collar of Tyr residues between α-ring subunits stabilizing the open gate conformation. **e**, Top-down view of Structure 4 and yeast pre-15S structure

(PDB:7LS6) shows the N-terminus of PAC1 (red) and Pba1 (hot pink) interacting with α-ring subunits. **f**, Close-up view of Structure 4 and yeast pre-15S structure (PDB:7LS6) shows the N-terminus of PAC1 (red) interacting with α1 N-terminus and Pba1 (hot pink) interacting with α2 N-terminus. **g**, Close-up view of Structure 4 and yeast pre-15S structure (PDB:7LS6) shows the N-terminus of PAC1 (red) and Pba1 (hot pink) making close contacts with POMP/Ump1, and β5-propeptide. Close-up cut view of proholo and premature maps shows PAC1 N-terminus visible after half CP fusion.

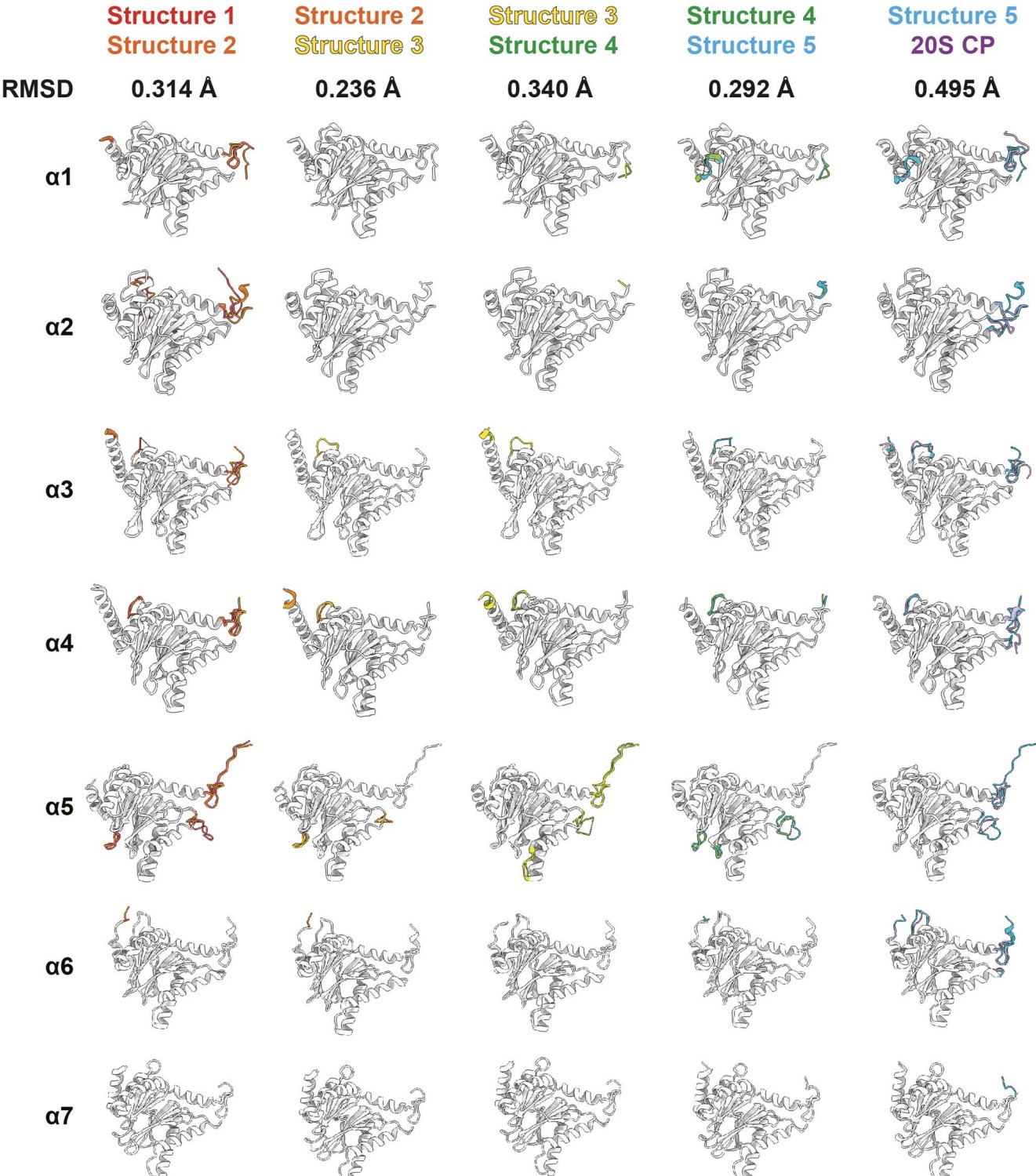

**Extended Data Fig. 6 | Structural comparison of α subunits across β-ring formation.** Pairwise structural comparison of α subunits between Structure 1 (red), Structure 2 (orange), Structure 3 (yellow), Structure 4 (green), Structure 5 (blue) and 20S CP (purple). Each subunit is individually aligned for each pair of comparison and only the difference between structures is colored.

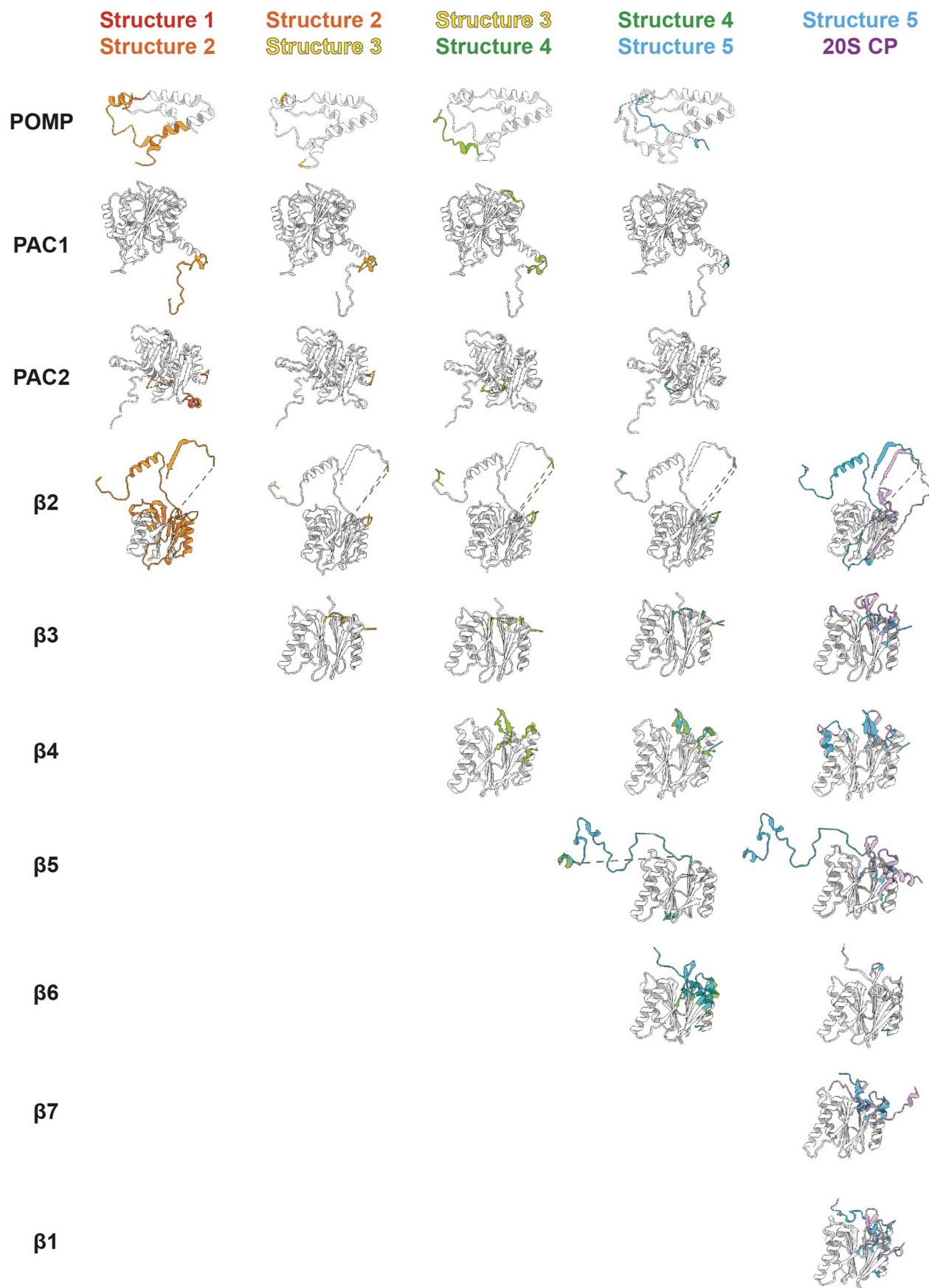

**Extended Data Fig. 7 | Structures of assembly chaperones and β subunits across β-ring formation.** Pairwise structural comparison of assembly factors POMP, PAC1/2 and β subunits between Structure 1 (red), Structure 2 (orange), Structure 3 (yellow), Structure 4 (green), Structure 5 (blue) and 20S CP (purple). Each subunit is individually aligned for each pair of comparison and only the difference between structures is colored.

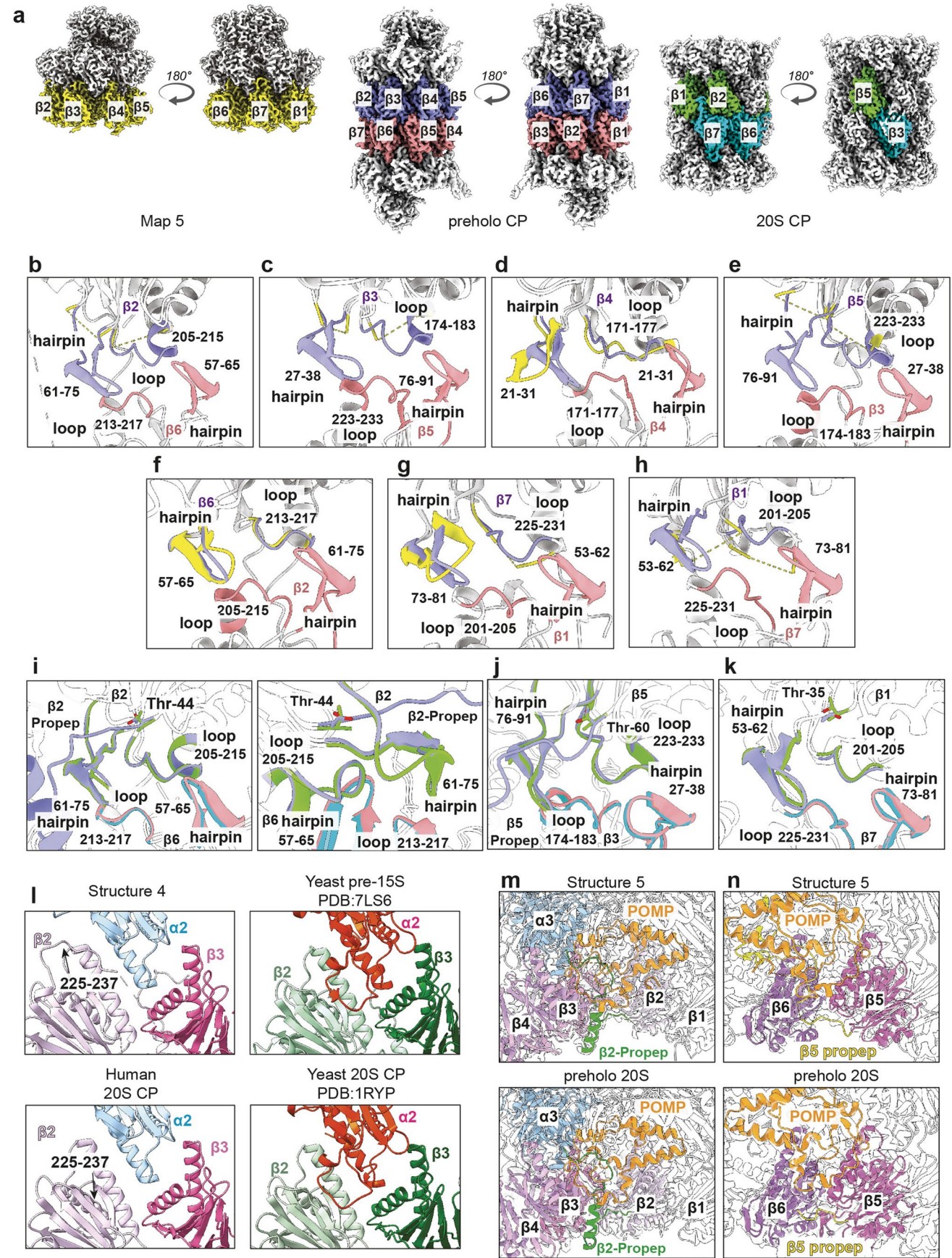

**Extended Data Fig. 8 | See next page for caption.**

**Extended Data Fig. 8 | Fusion-tetrad formed by the fusion hairpins and loops from opposing β subunits upon half CP fusion. a**, Cryo-EM maps of Structure 5, preholo CP, and mature 20S CP with β-subunits colored in yellow, in purple and salmon in the two halves, or in green and cyan in the two halves, respectively. β-subunit coloring is maintained in panels **b-k. b**, Close-up comparing fusion "hairpin" and "loop" in β2 between Structure 5 and preholo CP where these interlock with fusion "hairpin" and "loop" from β6 from the opposite half-CP. **c**, Close-up comparing fusion "hairpin" and "loop" in β3 between Structure 5 and preholo CP where these interlock with β5 from the opposite half-CP. **d**, Close-up comparing fusion "hairpin" and "loop" in β4 between Structure 5 and preholo CP where these interlock with β4 from the opposite half-CP. **e**, Close-up comparing fusion "hairpin" and "loop" in β5 between Structure 5 and preholo CP where these interlock with β3 from the opposite half-CP. **f**, Close-up comparing fusion "hairpin" and "loop" in β6 between Structure 5 and preholo CP where these interlock with β2 from the opposite half-CP. **g**, Close-up comparing

fusion "hairpin" and "loop" in β7 between Structure 5 and preholo CP where these interlock with β1 from the opposite half-CP. **h**, Close-up comparing fusion "hairpin" and "loop" in β1 between Structure 5 and preholo CP where these interlock with β7 from the opposite half-CP. **i**, Close-up comparing preholo CP and mature 20S CP fusion "hairpins" and "loops" interlocking between β2 in one half CP and β6 from the opposite half. **j**, Close-up comparing preholo CP and mature 20S CP fusion "hairpins" and "loops" interlocking between β5 in one half CP and β3 from the opposite half. **k**, Close-up comparing preholo CP and mature 20S CP fusion "hairpins" and "loops" interlocking between β1 in one half CP and β7 from the opposite half. **l**, Close-ups showing distinct α2 loop and trajectory of β2 C-terminal loop between human and yeast, as shown in Structure 4 and yeast pre-15S (PDB: 7LS6), respectively. **m**, Close-ups comparing β2-propeptide interactions in Structure 5 and in preholo-20S CP. **n**, Close-ups comparing β5-propeptide interactions in Structure 5 and preholo-20S CP.

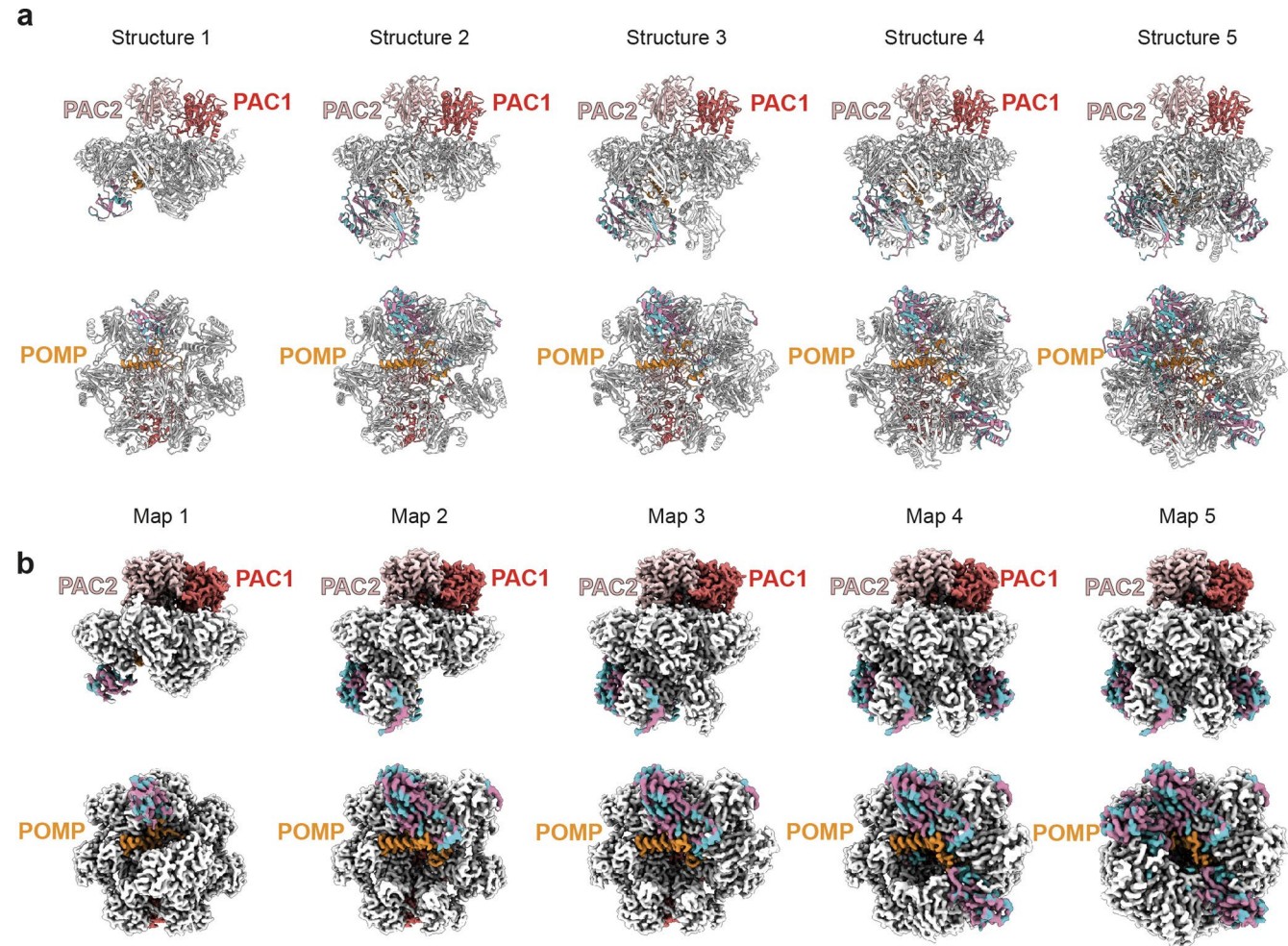

**Extended Data Fig. 9 | Sequence conservation between constitutive proteasome and immunoproteasome subunits mapped on Structures and Maps 1 to 5. a**, Structures 1 through 5 are shown from left to right in two views. β2, β5, and β1 subunits are colored based on sequence conservation with their counterparts in the immunoproteasome (sequence identity in maroon, divergence in cyan). PAC1/2 and POMP are colored brick red, salmon, and orange, respectively. **b**, Same as **a**, except showing the cryo-EM maps.

Frank Adolf
John Hanna

# Reporting Summary

## Statistics

For all statistical analyses, confirm that the following items are present in the figure legend, table legend, main text, or Methods section.

| n/a | Confirmed | |
|---|---|---|
| ☐ | ☒ | The exact sample size (*n*) for each experimental group/condition, given as a discrete number and unit of measurement |
| ☐ | ☒ | A statement on whether measurements were taken from distinct samples or whether the same sample was measured repeatedly |
| ☒ | ☐ | The statistical test(s) used AND whether they are one- or two-sided<br>*Only common tests should be described solely by name; describe more complex techniques in the Methods section.* |
| ☒ | ☐ | A description of all covariates tested |
| ☒ | ☐ | A description of any assumptions or corrections, such as tests of normality and adjustment for multiple comparisons |
| ☒ | ☐ | A full description of the statistical parameters including central tendency (e.g. means) or other basic estimates (e.g. regression coefficient) AND variation (e.g. standard deviation) or associated estimates of uncertainty (e.g. confidence intervals) |
| ☒ | ☐ | For null hypothesis testing, the test statistic (e.g. *F*, *t*, *r*) with confidence intervals, effect sizes, degrees of freedom and *P* value noted<br>*Give P values as exact values whenever suitable.* |
| ☒ | ☐ | For Bayesian analysis, information on the choice of priors and Markov chain Monte Carlo settings |
| ☒ | ☐ | For hierarchical and complex designs, identification of the appropriate level for tests and full reporting of outcomes |
| ☒ | ☐ | Estimates of effect sizes (e.g. Cohen's *d*, Pearson's *r*), indicating how they were calculated |

*Our web collection on statistics for biologists contains articles on many of the points above.*

## Software and code

Policy information about availability of computer code

| | |
|---|---|
| Data collection | Cryo-EM: SerialEM v4.1; Gel imaging: Amersham Imager 600 |
| Data analysis | Cryo-EM: CryoSparc v4.2.0;  Structure Analysis and Visualization: Chimera v1.15, ChimeraX v1.4, DeepEMhancer (https://github.com/rsanchezgarc/deepEMhancer); Model Building: COOT v0.9.6, Phenix.refine v1.19.2-4158, AlphaFold2, ISOLDE 1.4, Clustal Omega |

For manuscripts utilizing custom algorithms or software that are central to the research but not yet described in published literature, software must be made available to editors and reviewers. We strongly encourage code deposition in a community repository (e.g. GitHub). See the Nature Portfolio guidelines for submitting code & software for further information.

## Data

Policy information about availability of data

All manuscripts must include a data availability statement. This statement should provide the following information, where applicable:
- Accession codes, unique identifiers, or web links for publicly available datasets
- A description of any restrictions on data availability
- For clinical datasets or third party data, please ensure that the statement adheres to our policy

The atomic coordinates and electron microscopy maps have been deposited in the PDB with accession code 8QYJ, 8QYL, 8QYM, 8QZ9, 8QYN, 8QYS and 8QYO and in the Electron Microscopy Data Bank with codes EMD-18755 for 20S assembly intermediate map 1, EMD-18757 for 20S assembly intermediate map 2, EMD-18758 for 20S assembly intermediate map 3, EMD-18773 for 20S assembly intermediate map 4, EMD-18759 for 20S assembly intermediate map 5, EMD-18761 for

preholo-20S core particle, EMD-18762 for premature-20S core particle, EMD-18760 for mature 20S core particle, EMD-19342 for beta 7 tagged half 20S core particle and EMD-19343 for beta 7 tagged mature 20S core particle. All other reagents and data (for example, raw gels of experiments and raw movie electron microscopy data) are available from the corresponding authors upon request.

# Research involving human participants, their data, or biological material

Policy information about studies with <u>human participants or human data</u>. See also policy information about <u>sex, gender (identity/presentation), and sexual orientation</u> and <u>race, ethnicity and racism</u>.

| | |
|---|---|
| Reporting on sex and gender | No research involving human participants has been performed |
| Reporting on race, ethnicity, or other socially relevant groupings | No research involving human participants has been performed |
| Population characteristics | No research involving human participants has been performed |
| Recruitment | No research involving human participants has been performed |
| Ethics oversight | No research involving human participants has been performed |

Note that full information on the approval of the study protocol must also be provided in the manuscript.

# Field-specific reporting

Please select the one below that is the best fit for your research. If you are not sure, read the appropriate sections before making your selection.

☒ Life sciences ☐ Behavioural & social sciences ☐ Ecological, evolutionary & environmental sciences

For a reference copy of the document with all sections, see <u>nature.com/documents/nr-reporting-summary-flat.pdf</u>

# Life sciences study design

All studies must disclose on these points even when the disclosure is negative.

| | |
|---|---|
| Sample size | Sample size calculations were not preformed. |
| Data exclusions | No data were excluded. |
| Replication | Three indpendent replicates were carried out for each purification. All attempts of replication were successful. |
| Randomization | No grouped samples. |
| Blinding | No grouped samples. |

# Reporting for specific materials, systems and methods

We require information from authors about some types of materials, experimental systems and methods used in many studies. Here, indicate whether each material, system or method listed is relevant to your study. If you are not sure if a list item applies to your research, read the appropriate section before selecting a response.

## Materials & experimental systems

| n/a | Involved in the study |
|---|---|
| ☒ | ☐ Antibodies |
| ☐ | ☒ Eukaryotic cell lines |
| ☒ | ☐ Palaeontology and archaeology |
| ☒ | ☐ Animals and other organisms |
| ☒ | ☐ Clinical data |
| ☒ | ☐ Dual use research of concern |
| ☒ | ☐ Plants |

## Methods

| n/a | Involved in the study |
|---|---|
| ☒ | ☐ ChIP-seq |
| ☒ | ☐ Flow cytometry |
| ☒ | ☐ MRI-based neuroimaging |

# Eukaryotic cell lines

Policy information about cell lines and Sex and Gender in Research

| | |
|---|---|
| Cell line source(s) | High five cell (BTI-TN-5B1-4) were obtained from ThermoFisher Scentific (catalogue number:B85502). Gibco Sf9 cells were obtained ThermoFisher Scentific (catalogue number:11496016). |
| Authentication | Cell lines were not authenticated. |
| Mycoplasma contamination | Cell lines were periodically tested for mycoplasm contamination with no contamination detected. |
| Commonly misidentified lines (See ICLAC register) | No commonly misidentified cell lines were used in this study. |

