## [Peer Review File · Nature Structural & Molecular Biology]

Peer Review Information

Manuscript Title: Visualizing chaperone-mediated multistep assembly of the human 20S proteasome

Corresponding author name(s): Brenda Schulman, John Hanna, Frank Adolf

Reviewer Comments & Decisions:

Decision Letter, initial version:
--

Message: 13th Dec 2023

Dear Professor Schulman,

Thank you again for submitting your manuscript "Visualizing chaperone-mediated multistep assembly of the human 20S proteasome". We now have comments (below) from the 3 reviewers who evaluated your paper. In light of these reports, we remain interested in your study and would like to see your response to the comments of the referees, in the form of a revised manuscript.

You will see that all experts appreciate the novelty and importance of the novel findings. However, all three reviewers additionally raise fairly similar concerns with respect to lack of clarity in the figures and text, missing technical information that is necessary to adequately assess the underlying data, and the need for further discussion and contextualisation of the data within existing literature. We thus ask you to please improve the clarity of the presentation in accordance with the guidance of the experts, address all concerns about additional information needed, and adequately discuss and/or introduce possible mechanistic implications of the findings vis-à-vis existing literature in a revised manuscript.

Please be sure to address/respond to all concerns of the referees in full in a point-by-point response and highlight all changes in the revised manuscript text file. If you have comments that are intended for editors only, please include those in a separate cover letter.

We expect to see your revised manuscript within 3 months. If you cannot send it within this time, please contact us to discuss an extension; we would still consider your revision, provided that no similar work has been accepted for publication at NSMB or published elsewhere.

Reporting Summary:

When submitting the revised version of your manuscript, please pay close attention to our [href="https://www.nature.com/nature-portfolio/editorial-policies/image-integrity">Digital Image Integrity Guidelines](https://www.nature.com/nature-portfolio/editorial-policies/image-integrity). and to the following points below:

If there are additional or modified structures presented in the final revision, please submit the corresponding PDB validation reports and please deposit maps/coordinates so that the referees may assess them.

Data availability: this journal strongly supports public availability of data. All data used in accepted papers should be available via a public data repository, or alternatively, as

Supplementary Information. If data can only be shared on request, please explain why in your Data Availability Statement, and also in the correspondence with your editor. Please note that for some data types, deposition in a public repository is mandatory - more information on our data deposition policies and available repositories can be found below: <https://www.nature.com/nature-research/editorial-policies/reporting-standards#availability-of-data>

[redacted]

Sincerely,

Dimitris Typas
Associate Editor
Nature Structural & Molecular Biology
ORCID: 0000-0002-8737-1319

Reviewers' Comments:

Reviewer #1:

Remarks to the Author:

Adolf et al. use cryoelectron microscopy to determine and characterize seven recombinant human proteasome core particle (CP) subcomplexes containing chaperones PAC1/PAC2, PAC3/PAC4 and POMP. The structure of a CP intermediate containing chaperones PAC3/4 is solved to high resolution, and a series of additional CP intermediates with increasing number of β subunits also solved as well as two supra premature 20S structures. Fig 1C nicely summarizes the observed structures. This study advances our understanding of proteasome assembly and biogenesis, but several concerns/comments are listed below.

One general comment is that the figure referencing could be clearer, as in many cases multiple figures are referenced in bundle at the end of a sentence. This style requires a lot of effort from the reader to search where each effect is present in the image – for example this sentence fragment “PAC3/4 occupy the α 3- α 7 side of the ring and, interestingly, a portion of α 5 is situated at the PAC3/4 interface (Fig. 2b-c, Extended Data Fig. 3b).”

It might be worth speculating on why there's no premature-20 CP structure with double capping of the PAC1/2 heterodimers and POMP already released.

With β 7 tagging, it seems that no Structure 5/Map5 state was observed and the SEC fractions seem to show no chaperone-bound state. Is this truly the case and if so, does this have assembly implications?

p. 7: The authors write “with Phe positioned where Tyr sits in more conventional HbYX motifs (Fig. 2a)” but this is not really visible in the figure.

p. 7: ‘PAC4’ should probably be replaced by ‘PAC3’ in this sentence: “Interestingly, the same site in POMP bound by PAC4 is recognized later during assembly by the β 5 propeptide (Extended Data Fig. 3f).”

p. 8: This effect is not visible in the image: “other end inserted between PAC4, α 3 and α 4 (Fig. 2d).”

p. 8, last paragraph: “Residues 114-141 directly clamp β 3 against the α -ring through an interaction that is also buttressed by part of the β 2-propeptide (Fig. 2e, bottom panel).” The authors should label POMP 114-141 in Fig. 2e, bottom panel.

p. 8 and p. 10: There's a typo for ‘propeptide’

p. 9: The ‘collar of Tyr residues’ does not appear to be shown.

p. 12: The last part of the full paragraph doesn't reflect the referenced figures.

p. 12: “For the β 2 propeptide, these mirror those in Structure 5, and include interactions with β 3, β 4, α 3, and POMP (Fig. 5d). Meanwhile, 52-59 residues of the β 5 propeptide are observed retaining contacts with β 6 (Fig. 5e).” In Fig. 5e, the β 5 propeptide is not shown to contact β 6 - maybe Fig. 4 should be referenced here? Also, Fig. 5d does not show β 4, α 3, and POMP.

Fig. 5e: It'd be better to move the label "76-91" close to the β -sheet (colored in purple) as shown in Fig. 5d.

p. 30: "There are additional regions of density around β 2 in structure 1, β 3 in structure 2, β 4 in structure 3, β 5 in structure 4 and β 1/7 in structure 3." Should the second 'structure 3' be 'structure 5'? Is this additional density from other β subunits?

Reviewer #2:

Remarks to the Author:

Proteasomes are essential proteolytic enzymes for maintenance of cellular homeostasis. It has been demonstrated that the assembly of the 20S core particle (20S CP), the catalytic center of the proteasome, is carried out stepwise with the assistance of five proteasome-specific assembly chaperones, PAC1-PAC4 (Pba1-Pba4 in yeast), and POMP (Ump1). Besides, the N-terminus propeptides of the β -subunits have been identified as an essential element for accurate assembly.

Previous studies have revealed the structures of the yeast assembly chaperones (Ump1, Pba1 and Pba2) and proteasome intermediates (13S, pre-15S), as well as the several structures of the individual chaperones with their interacting subunits. In this manuscript, Adolf et al. employed a cryo-EM study to reveal the sequential snapshots of the human proteasome assembly intermediates. These structures provide a clear depiction of the chaperone-mediated assembly process leading to the accurate holocomplex. The data beautifully demonstrate the incremental integration of the β subunits into the assembly intermediates with assistance of the chaperones; particularly, distinct segments of POMP are sequentially replaced by corresponding β subunits. Notably, the Pre-holo complex, and early intermediates between the α -ring and chaperone complexes, are newly reported, providing new insights into the assembly process. The visualization of the process of the regulated assembly is compelling. My major concern is most of the figures do not clearly convey their message. I recommend the publication of a revised version in Nature Structural and Molecular Biology. The following are my concerns/comments.

1. In the papers by Hirano et al. in 2005 and 2006, which unveiled the proteasome assembly chaperones, RNAi was used to accumulate intermediates. Detecting assembly intermediates under normal conditions proves challenging due to the presumably fast kinetics of the assembly. However, in the pBac-based expression system, these assembly intermediates were accumulated. The reviewer raises a question regarding the inefficiency of the assembly process in their system in comparison with cellular condition. It would be great to discuss why the assembly is more efficient in the cells.
2. They described that the N-terminal tails of the α subunits alter the conformation. However, those changes are difficult to understand from the figures. Please present a figure indicating the change.
3. While the α -ring is consistently present in all intermediates, it remains unclear if there are any conformational changes in the α -subunit aside from the N-terminal tails. It would be beneficial to include Residual RMSD data for visualizing these potential differences.
4. Please illustrate the changes in the position of the incorporated β -subunits compared to that in the holocomplex.
5. The accuracy of modeling the flexible segments such as propeptides of the β -subunits, POMP1, and the N-terminus of the α -subunit in the individual EM maps remains unclear. To address this, please provide the EM density to validate the presented models.
6. Please provide a detailed description of the conformational change in the N-terminus of

the α -subunit upon insertion of the HbYX motifs.

7. Extended data Figures 5 and 6 lack a legend. Please include a legend for clarify.

8. Is the structure of POMP different from the yeast Ump1?

9. Please describe more details of the process from the preholo complex to the mature complex. It would be nice to discuss about the eviction of PAC1 and PAC2 from the mature Form.

10. On page14 of the discussion, there is a description of Immunoproteasome assembly. Further exploration and discussion on sequence alignment, and postential preference in incorporation would enhance the understanding.

11. Citation in page 4 line 7: Ramos et al., Cell, 1998 Li et al., EMBO 2007 for the Ump1 function

12. Citation in page 4 line 6: Murata et al., 2009 for the 20S assembly pathway

Reviewer #3:

Remarks to the Author:

Adolf et al. visualized a comprehensive series of CP assembly intermediates (mainly focusing on beta subunit additions) with all 5 assembly chaperones and active site propeptides, using recombinant system and cryo-EM reconstructions. Their findings explain how chaperones act together to order beta subunit addition, and visualize intriguing conformational changes in both the CP subunits and chaperones along the progression of the assembly events, which enable maturation of the CP active sites. The data are presented in a logical manner. This reviewer is enthusiastic about the findings, and suggests several changes to enhance the clarity of the manuscript.

1. In Fig. 1C, the authors should provide some quantitative information, as to what % of each distinct structure is observed (map 1-5) during cryo-EM of the pooled fractions from Fig. 1a. This study captured some of the scarce assembly intermediates, which could be only observed using a specific mutant when endogenous CPs are analyzed. It would be informative to know whether this advantage is simply due to increased production of all subunits in the recombinant system or whether the relative abundance also differs among assembly intermediates in this system.

2. It should be defined in the figure 1C legend what X marks (subunit absent?) vs. check marks indicate.

3. Fig. 1C legend states that asterisk indicates "novel" bands in the later fractions that likely correspond to the assembly chaperones, PAC1-4 and POMP. These bands are not "novel" since all 5 chaperones have been previously identified and characterized. This sentence should be rephrased.

4. In Fig 2h description of structure 3, the authors state 2 differences between yeast and human CP structures. #1) alpha2 N-terminus in human CP (not alpha1 as in yeast), enters the CP interior. The authors should discuss some basis for this difference, including the positioning/length of alpha2 and alpha1 N-terminus relative to Pac1. #2) The authors should clarify whether PAC1 N-terminus also contacts the beta5-propeptide in human CP, as in yeast.

5. In Discussion, p13, "PAC3/4 may help to recruit POMP through direct interaction..." However, the observed contacts between PAC3/4 & POMP could be consequential, not causal, since both chaperones individually bind to the alpha ring. The authors should provide experimental evidence or reference for this claim; for example, without PAC3/4, POMP binding to the alpha ring should be disrupted.

6. On p28, in Cloning, Protein Expression, and Purification section, complete the information where Addgene plasmids are indicated as "#XXXX".

Author Rebuttal to Initial comments

Point-by-point Responses to Reviewers' Comments:

Reviewer #1:

Remarks to the Author:

Adolf et al. use cryoelectron microscopy to determine and characterize seven recombinant human proteasome core particle (CP) subcomplexes containing chaperones PAC1/PAC2, PAC3/PAC4 and POMP. The structure of a CP intermediate containing chaperones PAC3/4 is solved to high resolution, and a series of additional CP intermediates with increasing number of β subunits also solved as well as two supra premature 20S structures. Fig 1C nicely summarizes the observed structures. This study advances our understanding of proteasome assembly and biogenesis,

We thank the reviewer for such kind comments and enthusiasm for our study!

but several concerns/comments are listed below.

One general comment is that the figure referencing could be clearer, as in many cases multiple figures are referenced in bundle at the end of a sentence. This style requires a lot of effort from the reader to search where each effect is present in the image – for example this sentence fragment “PAC3/4 occupy the α 3- α 7 side of the ring and, interestingly, a portion of α 5 is situated at the PAC3/4 interface (Fig. 2b-c, Extended Data Fig. 3b).”

We addressed this in two ways. First, to improve visualizing key features of the structures, we have rearranged and added figure panels. We also edited Supplementary videos 1-5, and added Supplementary videos 6-10, to better visualize the step-by-step structural progression. Second, we carefully redid the figure callouts throughout the text.

In the revised manuscript, this particular portion of the next now reads:

PAC3/4 sits on the opposite face of the α -ring (i.e., the site of the future β -ring) (Fig. 2a-b, Supplementary videos 1 and 6). Consistent with a prior crystal structure of the orthologous isolated yeast Pba3/4- α 5 complex (Extended Data Fig. 5a)^{47,67,68}, PAC3/4 occupy the α 3- α 7 side of the ring, with a portion of α 5 situated at the PAC3/4 interface (Fig. 2b). PAC3/4's position on the α -ring corresponds to that of the β 4- β 7 subunits later during assembly (Supplementary video 6, bottom view). By occupying these sites, the PAC3/4 heterodimer not only stabilizes the α -ring to provide a platform for β -ring assembly, but also precludes incorporation of the late β -subunits, helping to order β -subunit incorporation.

It might be worth speculating on why there's no premature-20 CP structure with double capping of the PAC1/2 heterodimers and POMP already released.

To address this, we added the following to the Results:

Although the human preholo-20S CP is 2-fold symmetric, and maintains much of the assembly factor scaffold, only a single PAC1/2 heterodimer is visible in the premature complex (Fig. 5a). Notably, this premature configuration was proposed in the original description of PAC1/2-mediated assembly³⁵, and observed, albeit at >20 Å resolution, for yeast⁶³. We presume that an intermediate with double-capping of the 20S CP by PAC1/2 and with POMP already released exists but was not observed. We speculate that the combination of relatively lower abundance than the pre-holo intermediate, and larger size than the mature 20S CP, limited the number of particles in the chromatography fraction we examined by cryo-EM. Nonetheless, persistence of at least one PAC1/2 heterodimer in the pre-mature-20S CP suggests that PAC1/2 elimination from the complex follows POMP degradation.

With β 7 tagging, it seems that no Structure 5/Map5 state was observed and the SEC fractions seem to show no chaperone-bound state. Is this truly the case and if so, does this have assembly implications?

To address this, we collected cryo-EM data on the peak fraction from the purification using the tagged $\beta 7$. As predicted by the reviewer, both the mature 20S and the Structure 5/Map5 state were observed. We added this information as Figure 1c, Extended Data Figure 2, and Supplementary Table 1. We also *discussed this in the Methods as follows*.

Importantly, the cryo-EM data showed that the peak fractions from both the complex purified based on the $\beta 2$ tag, and the complex based on the $\beta 7$ tag, were dominated by fully-assembled, mature 20S CPs (see Extended Data Figs. 1 and 2). In accordance with the expression and purification strategy, Map 5 (termed 5⁷ for the $\beta 7$ -tagged complex) was refined for both purifications. Map 5 for the $\beta 2$ -tagged complex was superior, presumably due to the purification enriching for complexes in this molecular weight range, and quality and quantity of data collected on the Titan Krios. Meanwhile, the refinement of Maps 1-4 correlates with the purification strategy based on the $\beta 2$ -tag and enrichment of complexes in their molecular weight range.

p. 7: The authors write “with Phe positioned where Tyr sits in more conventional HbYX motifs (Fig. 2a)” but this is not really visible in the figure.

We apologize for lack of clarity. In response to this suggestion, and a related comment from Reviewer #2, we now show the motifs from PAC1/2 side-by-side with a conventional HbYX motifs from the 26S Proteasome in Figure 2d-f.

p. 7: ‘PAC4’ should probably be replaced by ‘PAC3’ in this sentence: “Interestingly, the same site in POMP bound by PAC4 is recognized later during assembly by the $\beta 5$ propeptide (Extended Data Fig. 3f).”

Thank you! We fixed the sentence.

p. 8: This effect is not visible in the image: “other end inserted between PAC4, $\alpha 3$ and $\alpha 4$ (Fig. 2d).”

We fixed the Figure (now Fig. 2h and i) to show this.

p. 8, last paragraph: “Residues 114-141 directly clamp $\beta 3$ against the α -ring through an interaction that is also buttressed by part of the $\beta 2$ -propeptide (Fig. 2e, bottom panel).” The authors should label POMP 114-141 in Fig. 2e, bottom panel.

We added the residue numbers to the figure (now Fig. 2j, bottom panel) for clarity.

p. 8 and p. 10: There’s a typo for ‘propeptide’

Thank you! We fixed the typos.

p. 9: The ‘collar of Tyr residues’ does not appear to be shown.

We now show the collar of Tyr residues in Extended Data Fig. 5d.

p. 12: The last part of the full paragraph doesn’t reflect the referenced figures.

p. 12: “For the $\beta 2$ propeptide, these mirror those in Structure 5, and include interactions with $\beta 3$, $\beta 4$, $\alpha 3$, and POMP (Fig. 5d). Meanwhile, 52-59 residues of the $\beta 5$ propeptide are observed retaining contacts with $\beta 6$ (Fig. 5e).” In Fig. 5e, the $\beta 5$ propeptide is not shown to contact $\beta 6$ - maybe Fig. 4 should be referenced here? Also, Fig. 5d does not show $\beta 4$, $\alpha 3$, and POMP.

We apologize for lack of clarity and now show the details as follows:

For the $\beta 2$ -propeptide, these mirror those in Structure 5, and include interactions with $\beta 3$, $\beta 4$, $\alpha 3$, and POMP (Extended Data Fig. 8m). Meanwhile, residues 52-59 of the $\beta 5$ propeptide are observed retaining contacts with $\beta 6$ (Fig. 5d). Second, the propeptides align their active site and adjacent residues critical for their cleavage⁵³ (Fig. 5c-e).

Fig. 5e: It'd be better to move the label "76-91" close to the β -sheet (colored in purple) as shown in Fig. 5d.

We moved the label as suggested.

p. 30: "There are additional regions of density around β 2 in structure 1, β 3 in structure 2, β 4 in structure 3, β 5 in structure 4 and β 1/7 in structure 3." Should the second 'structure 3' be 'structure 5'? Is this additional density from other β subunits?

We fixed the typo and added a sentence for clarity:

There are additional regions of density around β 2 in structure 1, β 3 in structure 2, β 4 in structure 3, β 5 in structure 4 and β 1/7 in structure 5, for which coordinates were not modeled due to ambiguity of these regions. The unmodeled density presumably corresponds to regions of these subunits and/or their neighbors that become ordered upon additional interactions formed during assembly.

Reviewer #2:

Remarks to the Author:

Proteasomes are essential proteolytic enzymes for maintenance of cellular homeostasis. It has been demonstrated that the assembly of the 20S core particle (20S CP), the catalytic center of the proteasome, is carried out stepwise with the assistance of five proteasome-specific assembly chaperones, PAC1-PAC4 (Pba1-Pba4 in yeast), and POMP (Ump1). Besides, the N-terminus propeptides of the β -subunits have been identified as an essential element for accurate assembly. Previous studies have revealed the structures of the yeast assembly chaperones (Ump1, Pba1 and Pba2) and proteasome intermediates (13S, pre-15S), as well as the several structures of the individual chaperones with their interacting subunits. In this manuscript, Adolf et al. employed a cryo-EM study to reveal the sequential snapshots of the human proteasome assembly intermediates. These structures provide a clear depiction of the chaperone-mediated assembly process leading to the accurate holocomplex. The data beautifully demonstrate the incremental integration of the β subunits into the assembly intermediates with assistance of the chaperones; particularly, distinct segments of POMP are sequentially replaced by corresponding β subunits. Notably, the Pre-holo complex, and early intermediates between the α -ring and chaperone complexes, are newly reported, providing new insights into the assembly process. The visualization of the process of the regulated assembly is compelling. My major concern is most of the figures do not clearly convey their message. I recommend the publication of a revised version in Nature Structural and Molecular Biology.

We thank the reviewer for such kind comments and enthusiasm for our study!

The following are my concerns/comments.

1. In the papers by Hirano et al. in 2005 and 2006, which unveiled the proteasome assembly chaperones, RNAi was used to accumulate intermediates. Detecting assembly intermediates under normal conditions proves challenging due to the presumably fast kinetics of the assembly. However, in the pBac-based expression system, these assembly intermediates were accumulated. The reviewer raises a question regarding the inefficiency of the assembly process in their system in comparison with cellular condition. It would be great to discuss why the assembly is more efficient in the cells.

To address this, and a related question from Reviewer #3, we added two new paragraphs to the Methods. The first is new subsection on considerations for the strategy, and the second was added to the section on structure determination.

To date, structural knowledge of chaperone-bound eukaryotic 20S CP assembly intermediates has largely relied on strategies applied to budding yeast^{47,49,56,63,64}, including: use of mutant strains that stall assembly and thus increase cellular abundance of such intermediates; affinity tagging a chaperone (Ump1) to purify its associated proteins; mixing separately purified chaperone and CP complexes; and co-expression and purification of subcomplexes for X-ray crystallography. Meanwhile, it would seem that the roughly 10-fold lower cellular concentrations of assembly

chaperones compared to α - and β -subunits⁷², and purification procedures enriching for fully-assembled complexes, would pose challenges for structurally characterizing the intermediates along human 20S CP assembly. However, ever since the first reports of PAC1/2 and PAC3, there was evidence of intermediate chaperone assemblies in mammalian cells^{35,36}. More recent proteome-wide interactome studies performed by IP-MS of C-terminally tagged β -subunits indicates interaction with not only other stable proteasome components, but also the chaperones PAC1-4 (PSMG1-4) and POMP. Endogenously-tagged 20S CP components were shown to interact not only with proteasome subunits, but also with chaperones at significantly lower stoichiometry⁷². Moreover, purifications based on exogenously-expressed affinity tagged β 2 (PSMB7) - but not β 7 (PSMB4) - showed the early-departing assembly factors PAC3 and PAC4 (PSMG3-4). Thus, it stands to reason that the challenges to enriching human 20S CP intermediates could in principle be overcome by (1) increasing the expression of assembly chaperones, (2) affinity purification based on tagging the first β -subunit (β 2) to be incorporated, and (3) size fractionation such that complexes within a sample would be of relatively more similar mass. To achieve this, we utilized systems for high-level expression of protein complexes in insect cells.

and

Importantly, the cryo-EM data showed that the peak fractions from both the complex purified based on the β 2 tag, and the complex based on the β 7 tags were dominated by fully-assembled, mature 20S CPs (see Extended Data Figs. 1 and 2). In accordance with the expression and purification strategy, Map 5 (termed 5⁷ for the β 7-tagged complex) was refined for both purifications. Map 5 for the β 2-tagged complex was superior, presumably due to the enrichment of lower molecular weight complexes, and quality and quantity of data collected on the Titan Krios. Meanwhile, the refinement of Maps 1-4 correlates with the purification strategy based on the β 2-tag and enrichment of lower molecular weight complexes.

2. They described that the N-terminal tails of the α subunits alter the conformation. However, those changes are difficult to understand from the figures. Please present a figure indicating the change.

To address this, we added Supplementary Video 7, which directly compares the conformations of the N-terminal tails of the α -subunits in Structure1, Structure5, the preholo-SCP and the mature CP, and also shows morphing between them to highlight the conformational differences.

3. While the α -ring is consistently present in all intermediates, it remains unclear if there are any conformational changes in the α -subunit aside from the N-terminal tails. It would be beneficial to include Residual RMSD data for visualizing these potential differences.

We added RMSD differences between structures to Extended Data Figure 6.

4. Please illustrate the changes in the position of the incorporated β -subunits compared to that in the holocomplex.

In our revised manuscript, Supplementary Video 6 shows the structural progression across assembly, including between Structure 5, the pre-holo intermediate, and the mature CP. Extended Data Figure 7 shows the progressive changes in individual β -subunits.

5. The accuracy of modeling the flexible segments such as propeptides of the β -subunits, POMP1, and the N-terminus of the α -subunit in the individual EM maps remains unclear. To address this, please provide the EM density to validate the presented models.

We apologize for lack of clarity on this issue in our original submission. In the revised manuscript, we now show EM density over the chaperones and β -subunit propeptides, and the N-termini of the α -subunits, in Extended Data Figure 4b-h. In the cover letter with our original submission, we had provided a link to the maps and models for confidential download, but we are not sure if this was conveyed to reviewers. We now provide the link both in the revised manuscript, and here: <https://datashare.biochem.mpg.de/s/QTzTIAIRzJCriep>.

6. Please provide a detailed description of the conformational change in the N-terminus of the α -subunit upon insertion of the HbYX motifs.

To address this, we added a description to the text (provided in the response to point #9 below from this reviewer), as well as Supplementary Video 10 comparing a complex with the PAC1/2 HbYX motifs engaged, and the mature CP.

7. Extended data Figures 5 and 6 lack a legend. Please include a legend for clarify.

Thank you! We have fixed this inadvertent omission from the original. The revised manuscript includes legends for all Figures, Extended Data Figures, Supplemental Figures, and Videos.

8. Is the structure of POMP different from the yeast Ump1?

The revised manuscript now shows superposition of POMP and yeast Ump1 as Extended Data Figure 5c.

9. Please describe more details of the process from the preholo complex to the mature complex. It would be nice to discuss about the eviction of PAC1 and PAC2 from the mature Form.

To address this, and point 6 from this reviewer, we added Supplementary videos 6-10 to better visualize the structural progression across β -ring assembly, half-CP fusion, and from the preholo complex to the mature complex. We also added another paragraph to the Results and to the Discussion, pasted below.

New paragraph in Results:

The preholo-, premature- and mature 20SCP complexes also show differences in the α -ring, which primarily impact gating and accessibility of the central pore (Supplementary video 10). The α -ring conformation is similar in Structures 2-5 and the preholo-20SCP (Supplementary video 7). These intermediates show common interactions with PAC1/2, POMP, the β 2-propeptide, and part of the β 5-propeptide. In these assemblies, the α -ring is largely occluded by elements coming from both directions (Extended Data Fig. 5g). The α 1 and PAC1 N-termini point from the outside towards the inside of the barrel. From the inside, POMP binds PAC1's N-terminus to further occupy the pore. These elements are not visible in the premature-20SCP, but the α -ring gate remains open: the majority of α -subunit N-termini are directed outward due engagement of PAC1/2. PAC1/2 thus poises the open gate conformation until the very end of CP maturation. The premature-20S CP would set the stage for the N-termini of α 2, α 3, α 4, and α 5 to fill the pore from the opposite side of PAC1/2's anchor, to adopt the closed gate conformation of the mature 20S CP (Supplementary video 10).

New paragraph in Discussion:

Meanwhile, the structural progression from pre-holo to premature to mature 20SCP shed light on final stages of assembly. As neither POMP nor the propeptides are visible in the premature-20S CP, cleavage of the propeptides might also destabilize the internal assembly factor scaffold, allowing for its degradation. Yet, maintenance of PAC1/2 could in principle facilitate exit of cleaved propeptides and POMP. It also seems that PAC1/2 binding would be destabilized by lack of contacts between PAC1's N-terminus with the α -ring and POMP. Future studies will be required to determine if PAC1/2 is degraded by its associated CP in cis, or in isolation after liberation from the mature CP.

10. On page 14 of the discussion, there is a description of Immunoproteasome assembly. Further exploration and discussion on sequence alignment, and postential preference in incorporation would enhance the understanding.

Although our original manuscript did not mention the immunoproteasome in the Discussion, to address this, we added the text below, and Extended Data Fig. 9 mapping immunoproteasome sequence differences on Structure 5.

Notably, differences between the canonical sequences used here and the immunoproteasome mapped on Structures 1-5 show sequence variations in several residues making contacts along the assembly process, including in propeptides, but that are exposed in the mature complex (Extended Data Fig. 9). Thus, future studies may reveal distinct features of assembly of specialized proteasomes.

11. Citation in page 4 line 7: Ramos et al., Cell, 1998 Li et al., EMBO 2007 for the Ump1 function

We apologize for these omissions from the original and added the citations.

12. Citation in page 4 line 6: Murata et al., 2009 for the 20S assembly pathway

We apologize for this omission from the original and added the citation.

Reviewer #3:

Remarks to the Author:

Adolf et al. visualized a comprehensive series of CP assembly intermediates (mainly focusing on beta subunit additions) with all 5 assembly chaperones and active site propeptides, using recombinant system and cryo-EM reconstructions. Their findings explain how chaperones act together to order beta subunit addition, and visualize intriguing conformational changes in both the CP subunits and chaperones along the progression of the assembly events, which enable maturation of the CP active sites. The data are presented in a logical manner. This reviewer is enthusiastic about the findings, and suggests several changes to enhance the clarity of the manuscript.

We thank the reviewer for such kind comments and enthusiasm for our study!

1. In Fig. 1C, the authors should provide some quantitative information, as to what % of each distinct structure is observed (map 1-5) during cryo-EM of the pooled fractions from Fig. 1a.

To address this, we have added the number of particles used for each reconstruction in Extended Data Fig. 4a.

This study captured some of the scarce assembly intermediates, which could be only observed using a specific mutant when endogenous CPs are analyzed. It would be informative to know whether this advantage is simply due to increased production of all subunits in the recombinant system or whether the relative abundance also differs among assembly intermediates in this system.

To address this, and a related question from Reviewer #2, we added two new paragraphs to the Methods. The first is new subsection on considerations for the strategy, and the second was added to the section on structure determination.

To date, structural knowledge of chaperone-bound eukaryotic 20S CP assembly intermediates has largely relied on strategies applied to budding yeast^{47,49,56,63,64}, including: use of mutant strains that stall assembly and thus increase cellular abundance of such intermediates; affinity tagging a chaperone (Ump1) to purify its associated proteins; mixing separately purified chaperone and CP complexes; and co-expression and purification of subcomplexes for X-ray crystallography. Meanwhile, it would seem that the roughly 10-fold lower cellular concentrations of assembly chaperones compared to α - and β -subunits⁷², and purification procedures enriching for fully-assembled complexes, would pose challenges for structurally characterizing the intermediates along human 20S CP assembly. However, ever since the first reports of PAC1/2 and PAC3, there was evidence of intermediate chaperone assemblies in mammalian cells^{35,36}. More recent proteome-wide interactome studies performed by IP-MS of C-terminally tagged β -subunits indicates interaction with not only other stable proteasome components, but also the chaperones PAC1-4 (PSMG1-4) and POMP. Endogenously-tagged 20S CP components were shown to interact not only

with proteasome subunits, but also with chaperones at significantly lower stoichiometry⁷². Moreover, purifications based on exogenously-expressed affinity tagged $\beta 2$ (PSMB7) - but not $\beta 7$ (PSMB4) - showed the early-departing assembly factors PAC3 and PAC4 (PSMG3-4). Thus, it stands to reason that the challenges to enriching human 20S CP intermediates could in principle be overcome by (1) increasing the expression of assembly chaperones, (2) affinity purification based on tagging the first β -subunit ($\beta 2$) to be incorporated, and (3) size fractionation such that complexes within a sample would be of relatively more similar mass. To achieve this, we utilized systems for high-level expression of protein complexes in insect cells.

and

Importantly, the cryo-EM data showed that the peak fractions from both the complex purified based on the $\beta 2$ tag, and the complex based on the $\beta 7$ tags were dominated by fully-assembled, mature 20S CPs (see Extended Data Figs. 1 and 2). In accordance with the expression and purification strategy, Map 5 (termed 5⁷ for the $\beta 7$ -tagged complex) was refined for both purifications. Map 5 for the $\beta 2$ -tagged complex was superior, presumably due to the enrichment of lower molecular weight complexes, and quality and quantity of data collected on the Titan Krios. Meanwhile, the refinement of Maps 1-4 correlates with the purification strategy based on the $\beta 2$ -tag and enrichment of lower molecular weight complexes.

2. It should be defined in the figure 1C legend what X marks (subunit absent?) vs. check marks indicate.

Thank you! We added this information to the legend:

A checkmark indicates that the protein was visualized in the complex. An "X" indicates the protein was not visualized in the complex.

3. Fig. 1C legend states that asterisk indicates "novel" bands in the later fractions that likely correspond to the assembly chaperones, PAC1-4 and POMP. These bands are not "novel" since all 5 chaperones have been previously identified and characterized. This sentence should be rephrased.

Thank you! We revised this sentence in the legend, as follows.

Asterisk indicates bands in the later fractions (corresponding to complexes smaller than a mature 20S CP) of lower molecular weight than proteasome subunits that would correspond to the assembly chaperones PAC1-4 and POMP.

4. In Fig 2h description of structure 3, the authors state 2 differences between yeast and human CP structures. #1) alpha1 N-terminus in human CP (not alpha2 as in yeast), enters the CP interior. The authors should discuss some basis for this difference, including the positioning/length of alpha2 and alpha1 N-terminus relative to Pac1. #2) The authors should clarify whether PAC1 N-terminus also contacts the beta5-propeptide in human CP, as in yeast.

We addressed this in several ways. First, we edited the text as pasted below.

Importantly, Structure 3 corresponds to the well-recognized 13S precursor^{39,55}, and is largely superimposable on the yeast 13S⁵⁶, although there are some notable differences accommodating the divergent N-terminal sequences and structures of PAC1 and yeast Pba1. First, while the N-termini of both PAC1 and Pba1 thread through the open gate into the CP interior, in yeast the N-terminus of $\alpha 2$ ⁶⁶, not $\alpha 1$, runs alongside Pba1's N-terminus (Extended Data Fig. 5e-f). Second, PAC1's longer N-terminus loops back upwards towards the α -ring gate whereas Pba1's ends in contact with Ump1 and the $\beta 5$ -propeptide⁵⁶ (Extended Data Fig. 5g).

Second, we remade the figure and added additional panels for clarity (now Extended Data Figs. 5e, f) to show the major sequence differences, which are in PAC1 and Pba1, in the context of the structures.

5. In Discussion, p13, "PAC3/4 may help to recruit POMP through direct interaction..." However, the observed contacts between PAC3/4 & POMP could be consequential, not causal, since both chaperones individually bind to the alpha ring. The authors should provide experimental evidence or reference for this claim; for example, without PAC3/4, POMP binding to the alpha ring should be disrupted.

We apologize for confusion with our initial wording. For accuracy, we have edited these sentences to:

First, PAC3/4 directly binds POMP. Second, PAC3/4 may help order β -subunit incorporation by preventing untimely incorporation of the later subunits through occlusion of their binding sites.

6. On p28, in Cloning, Protein Expression, and Purification section, complete the information where Addgene plasmids are indicated as "#XXXX".

We added the codes for the Addgene plasmid submissions ((# 214137, #214138, #214139, #214140).

Decision Letter, first revision:

Message: Our ref: NSMB-A48434A

26th Jan 2024

Dear Professor Schulman,

Thank you for submitting your revised manuscript "Visualizing chaperone-mediated multistep assembly of the human 20S proteasome" (NSMB-A48434A). It has now been seen by the original referees and their comments are below. The reviewers find that the paper has improved in revision, and therefore we are happy to accept it in principle in Nature Structural & Molecular Biology, pending minor revisions to satisfy the referees' final requests and to comply with our editorial and formatting guidelines.

We are now performing detailed checks on your paper and will send you a checklist detailing our editorial and formatting requirements in about two weeks. Please do not upload the final materials and make any revisions until you receive this additional information from us.

To facilitate our work at this stage, it is important that we have a copy of the main text as a word file. If you could please send along a word version of this file as soon as possible, we would greatly appreciate it; please make sure to copy the NSMB account (cc'ed above).

Sincerely,

Dimitris Typas
Associate Editor
Nature Structural & Molecular Biology
ORCID: 0000-0002-8737-1319

Reviewer #1 (Remarks to the Author):

All my concerns have been addressed; congratulations to the authors on this impressive manuscript.

Reviewer #2 (Remarks to the Author):

All of my comments have been addressed by the authors. The newly provided figures, especially the movies, help us understand the structural details.

Reviewer #3 (Remarks to the Author):

Authors sufficiently addressed all of my comments on the initial version of the manuscript.

Author Rebuttal, first revision:

Responses to Reviewers in blue

Reviewer #1 (Remarks to the Author):

All my concerns have been addressed; congratulations to the authors on this impressive manuscript.

We thank the reviewer for such kind and thoughtful comments.

Reviewer #2 (Remarks to the Author):

All of my comments have been addressed by the authors. The newly provided figures, especially the movies, help us understand the structural details.

We thank the reviewer for such kind and thoughtful comments.

Reviewer #3 (Remarks to the Author):

Authors sufficiently addressed all of my comments on the initial version of the manuscript.

We thank the reviewer for appreciating our efforts at improving the revision.

Final Decision Letter:

Message: 6th Mar 2024

Dear Professor Schulman,

We are now happy to accept your revised paper "Visualizing chaperone-mediated multistep assembly of the human 20S proteasome" for publication as an Article in Nature Structural & Molecular Biology.

Acceptance is conditional on the manuscript's not being published elsewhere and on there

being no announcement of this work to the newspapers, magazines, radio or television until the publication date in Nature Structural & Molecular Biology.

Your paper will be published online soon after we receive proof corrections and will appear in print in the next available issue. You can find out your date of online publication by contacting the production team shortly after sending your proof corrections.

Please note that *Nature Structural & Molecular Biology* is a Transformative Journal (TJ). Authors may publish their research with us through the traditional subscription access route or make their paper immediately open access through payment of an article-processing charge (APC). Authors will not be required to make a final decision about access to their article until it has been accepted. Find out more about Transformative Journals

Authors may need to take specific actions to achieve compliance with funder and institutional open access mandates. If your research is supported by a funder that requires immediate open access (e.g. according to Plan S principles) then you should select the gold OA route, and we will direct you to the compliant route where possible. For authors selecting the subscription publication route, the journal's standard licensing terms will need to be accepted, including self-archiving policies. Those licensing terms will supersede any other terms that the author or any third party may assert apply to any version of the manuscript.

If you have any questions about our publishing options, costs, Open Access requirements,

or our legal forms, please contact ASJournals@springernature.com

Sincerely,

Dimitris Typas
Associate Editor
Nature Structural & Molecular Biology
ORCID: 0000-0002-8737-1319